# Investigating the differences in calculating global mean surface CO$_2$ abundance: the impact of analysis methodologies and site selection

**Zhendong Wu**[1,2]**, Alex Vermeulen**[2]**, Yousuke Sawa**[3]**, Ute Karstens**[1,2]**, Wouter Peters**[4,5]**, Remco de Kok**[1,4]**, Xin Lan**[6,7]**, Yasuyuki Nagai**[3]**, Akinori Ogi**[3]**, and Oksana Tarasova** TS1 [8]

[1]ICOS Carbon Portal, Department of Physical Geography and Ecosystem Sciences,
Lund University, Lund, Sweden
[2]ICOS ERIC, Carbon Portal, Lund, Sweden
[3]Japan Meteorological Agency (JMA), Tokyo, Japan
[4]Environmental Sciences Group, TS2 Wageningen University, Wageningen, the Netherlands
[5]Centre for Isotope Research, University of Groningen, Groningen, the Netherlands
[6]NOAA Global Monitoring Laboratory (GML), Boulder, CO, USA
[7]Cooperative Institute for Research in Environmental Sciences, University of Colorado, Boulder, CO, USA
[8]WMO, Geneva, Switzerland

**Correspondence:** Zhendong Wu (zhendong.wu@nateko.lu.se) and Alex Vermeulen
(alex.vermeulen@icos-ri.eu)

**Abstract.** The World Meteorological Organization (WMO) Global Atmosphere Watch (GAW) coordinates high-quality atmospheric greenhouse gas observations globally and provides these observations through the WMO World Data Centre for Greenhouse Gases (WDCGG) supported by Japan Meteorological Agency. The WDCGG and the National Oceanic and Atmospheric Administration (NOAA) analyse these measurements using different methodologies and site selection to calculate global annual mean surface CO$_2$ and its growth rate as a headline climate indicator. This study introduces a third hybrid method named GFIT, which serves as an independent validation and open-source alternative to the methods described by NOAA and WDCGG. We apply GFIT to incorporate observations from most WMO GAW stations and 3D modelled CO$_2$ fields from Carbon-Tracker Europe (CTE). We find that different observational networks (i.e. NOAA, GAW, and CTE networks) and analysis methods result in differences in the calculated global surface CO$_2$ mole fractions equivalent to the current atmospheric growth rate over a 3-month period. However, the CO$_2$ growth rate derived from these networks and the CTE model output shows good agreement. Over the long-term period (40 years), both networks with and without continental sites exhibit the same trend in the growth rate ($0.030 \pm 0.002$ ppm yr$^{-1}$ each year). However, a clear difference emerges in the short-term (1-month) change in the growth rate. The network that includes continental sites improves the early detection of changes in biogenic emissions.

## 1 Introduction

Global mean surface temperature averaged over 2011–2020 has increased by about 1.09 °C relative to the average temperature of 1850–1900 (Gulev et al., 2021). The increasing amount of atmospheric carbon dioxide ($CO_2$), together with increases in other greenhouse gases, is the main driver of the warming (Eyring et al., 2021). After being relatively stable between 180 ppm (ice age) and 280 ppm (interglacial) for the past 800 000 years (Lüthi et al., 2008), the annual average $CO_2$ level of the atmosphere has increased since the industrial revolution from roughly 277 ppm in 1750 to $415.7 \pm 0.2$ ppm in 2021 (WMO, 2022), due to emissions of $CO_2$ related to human activities like burning of fossil fuels and land use changes (Friedlingstein et al., 2022). Mean global atmospheric $CO_2$ annual growth rate ($G_{ATM}$) is an important constraint on the global carbon cycle. Based on the most recent global carbon budget (GCB) analysis (Friedlingstein et al., 2022), the total emission of $CO_2$ due to human activities was $10.2 \pm 0.8$ GtC yr$^{-1}$ in 2020, of which $3.0 \pm 0.4$ GtC yr$^{-1}$ was captured by the ocean sink and $2.9 \pm 1$ GtC yr$^{-1}$ by the terrestrial sink, leaving a net increase of $5.0 \pm 0.2$ GtC yr$^{-1}$ of $CO_2$ in the atmosphere, corresponding to an atmospheric $CO_2$ mole fraction increase of $2.4 \pm 0.1$ ppm yr$^{-1}$. (The conversion factor comes from Ballantyne et al., 2012.)

As the atmosphere mixes the contributions of all sources and sinks, an observational global average $CO_2$ mole fraction can be constructed if there are enough observations to represent the spatial and temporal variations across the globe. Since most land masses are concentrated in the Northern Hemisphere, and the highest anthropogenic emissions (e.g. during winter) occur in the relatively narrow latitudinal band between 30 and 60° N, relatively large spatial and temporal gradients in $CO_2$ mole fraction exist in and around that region. Due to convective and advective mixing, the average mixing time of air within the same latitudinal bands varies from several weeks to a month. However, mixing between latitudinal bands is slower, especially the exchange between the Northern and Southern hemispheres, which has an approximate interhemispheric transport time of $1.4 \pm 0.2$ years (Patra et al., 2011). The interplay of the latitudinal and interhemispheric differences in fossil fuel emissions and seasonal exchange with land biota (Denning et al., 1995) creates a latitudinal and interhemispheric gradient that requires a sufficiently dense network to capture a representative global annual mean.

However, measurement stations that are close to sources or sinks may not be representative of a large atmospheric volume and the average signal at their latitude. Therefore, inclusion of these observations might introduce biases on the global mean $CO_2$ and its growth rate. These biases can be avoided by filtering of data and a careful selection of spatially representative stations, as done by NOAA in their use of 43 stations (Fig. 1) that are considered to be representative for the Marine Boundary Layer (MBL reference network, https://www.esrl.noaa.gov/gmd/ccgg/mbl/mbl.html;last access: 7 December 2023 TS3). An additional data processing step developed by NOAA to further avoid biases due to unrepresentative local signals is filtering and smoothing, by using a combination of a low pass filter and decomposition into a fitted long-term trend and seasonal cycle (Thoning et al., 1989), hereafter referred to as the NOAA analysis. These fits can also be used to fill gaps for missing data, though care must be taken to avoid extrapolation errors before and beyond the time covered by the data record of the station. The WMO Global Atmosphere Watch (GAW) World Data Centre for Greenhouse Gases (WDCGG) publishes global averages mole fraction for $CO_2$ and the other major greenhouse gases in the annual WMO GAW Greenhouse Gas Bulletin (latest version: WMO, 2022). They use curve fitting and filter methods that are very similar to those developed by NOAA, but WDCGG includes continental locations that are potentially more influenced by local sources and sinks (Tsutsumi et al., 2009).

The NOAA MBL observations are all part of the NOAA cooperative global air sampling network and analysed in the same laboratory. All NOAA flask–air observations are traceable to the current scale WMO–$CO_2$–X2019 (Hall et al., 2021) that is maintained by NOAA Global Monitoring Laboratory (GML). In contrast, the WDCGG data originate from multiple independent laboratories (including NOAA GML), that together form a network of hundreds of stations coordinated by WMO GAW (http://gawsis.meteoswiss.ch; last access: 7 December 2023). Having a multitude of independent laboratories carries an additional risk of biases due to differences in sampling, measurement, and analysis methods, for example calibration scales, although much care is taken to avoid these by coordination in the network and use of a common calibration scale from WMO Central Calibration Laboratories (CCL) guided by a set of strict measurement compatibility goals (WMO, 2022). The different selection of stations results in a larger seasonal cycle amplitude in WDCGG results compared with those of NOAA and a small but quite consistent bias in global surface annual mean $CO_2$ mole fraction (Tsutsumi et al., 2009). The NOAA estimate of global surface annual mean $CO_2$ mole fraction is expected to be lower (e.g. $\sim 0.35$ ppm lower than the WDCGG estimate, Tsutsumi et al., 2009) compared with a full global surface average because areas with large sources are not represented. However, the two aforementioned approaches neither represent the parts of the atmosphere with low $CO_2$ mole fraction levels (i.e. the full troposphere, up to $\sim 8$–15 km altitude, and the stratosphere), nor do they cover the regions of the world with substantial observational gaps.

In this paper, we propose a data integration method to estimate the global mean surface $CO_2$ and its growth rate, named GFIT. This method serves as an independent validation of the methods as described by NOAA and WDCGG through a completely independent and open-source implementation.

The global mean surface $CO_2$ refers to the mean $CO_2$ mole fraction within the planetary boundary layer, which extends from the earth's surface up to a few hundred or thousand metres in height. We apply the GFIT methodology to incorporate $CO_2$ data from the GAW network (139 stations; Fig. 1) and the modelled $CO_2$ distribution from a well-established 3D global transport model (TM5: Transport Model 5; Krol et al., 2005; Peters et al., 2004). We investigate the influence of small differences between the three methodologies and whether these are significant or not for calculating the global mean surface $CO_2$ and its growth rate, how consistent the GFIT and WDCGG approaches are with each other, and how they compare with NOAA analysis and estimates derived from a $CO_2$ simulation with the 3D transport model TM5. These 3D $CO_2$ results for 2001–2020 using TM5 are performed in the CarbonTracker Europe framework (CTE; Peters et al., 2004; Van Der Laan-Luijkx et al., 2017), where the $CO_2$ uptake and emission fluxes are optimized by the inversion system to minimize the mismatch between the in situ observations and the modelled $CO_2$ mole fraction. CTE generally has a good representation of the $CO_2$ field, with mean biases with respect to independent aircraft measurements of generally less than 0.5 ppm (Friedlingstein et al., 2022). Furthermore, the inferred $CO_2$ fluxes from CTE fit well within the ensemble of those of other inversions used for the evaluation of global carbon budget (e.g. Friedlingstein et al., 2022).

## 2 Methods and data

### 2.1 The WMO GAW observations and WDCGG analysis method

The WMO GAW network measurements are archived and distributed by WDCGG, hosted by the Japan Meteorological Agency. The GAW observations used in this study originate from 139 selected stations of the GAW network, and all observations are on the WMO standard scale WMO–$CO_2$–X2019. The details on the station selection are described in Tsutsumi et al. (2009), which mainly excludes stations located in the Northern Hemisphere that show large standard deviations from the latitudinal fitted curve. The remaining 139 stations show a more reasonable latitudinal scatter range (Fig. 1).

The WDCGG global analysis method (hereafter WDCGG method), as described in Tsutsumi et al. (2009), includes the mentioned station selection, a data fitting and filter (involves data interpolation and extrapolation), and calculation of the zonal and global mean mole fractions, trends, and growth rates. The procedure is also summarized in Sect. S1 in the Supplement. The output from the global analysis by the WDCGG method is used to compare with an alternative method (GFIT) that we designed to follow as closely as possible the fit and filter method (Conway et al., 1994) deployed by NOAA and is described in the Sect. 2.3.

### 2.2 CTE model output and station observations

CarbonTracker Europe is a global model of atmospheric $CO_2$ and designed to keep track of $CO_2$ uptake and release at the earth's surface over time (Van Der Laan-Luijkx et al., 2017). CTE incorporates an off-line atmospheric transport module (TM5, Peters et al., 2004; Krol et al., 2005) driven by ECMWF ERA5 data, and there are four prescribed fluxes (i.e. from ocean, biosphere, fire, and fossil fuel), which are transported in the model, together with the transported initial $CO_2$ field. CTE also includes a data assimilation system that applies an ensemble Kalman filter to optimize the biogenic and ocean fluxes for a combination of plant-functional types and climate zones to improve the fit of the simulated concentrations with observations. The optimized fluxes from the data assimilation have been used in Global Carbon Project (GCP) 2021, and the comparison of CTE $CO_2$ product with the other data assimilation systems used in GCP shows good agreement (within 0.8 ppm at all latitude bands) (Friedlingstein et al., 2022).

The CTE model data used here consist of simulated monthly $CO_2$ mole fraction at $1 \times 1°$ horizontal resolution and 25 levels in the vertical direction, and the data period ranges from 2001 to 2020 which has no influence of model spin-up (Krol et al., 2018). From the CTE output a set of simulated synthetic atmospheric $CO_2$ mole fractions with monthly resolution can be extracted within grid cells where stations are situated. This study analyses monthly observation data (1980–2020) and synthetic time series (2001–2020) by using the GFIT method (Sect. 2.3) and attempts to estimate global mean $CO_2$ mole fraction and its growth rate. The observed $CO_2$ mole fractions are taken from 230 out of 290 global-wide distributed stations (Fig. 1; the station selection is summarized in Sect. S2). The data come from the GLOBALVIEW-plus V8 ObsPack data product (Schuldt et al., 2022) and include surface-based, shipboard-based, and tower-based measurements.

### 2.3 The GFIT method

The temporal pattern of $CO_2$ measurement records at locations around the globe can be explained as the combination of roughly three components: a long-term trend, a non-sinusoidal yearly cycle (or seasonality), and short-term variations. This study synchronizes monthly $CO_2$ records with the fitting and filter method developed at the NOAA Global Monitoring Laboratory (Conway et al., 1994; Thoning et al., 1989), without extrapolation. The station selection and $CO_2$ averaging method are kept the same as in the WDCGG method (Sect. S1). This method will be referred to as the GFIT method and will be compared with the WDCGG method without extrapolation. The only difference from the WDCGG method without extrapolation is the fitting and filter method. All code for the method described here was developed in Python and is available as a Jupyter notebook un-

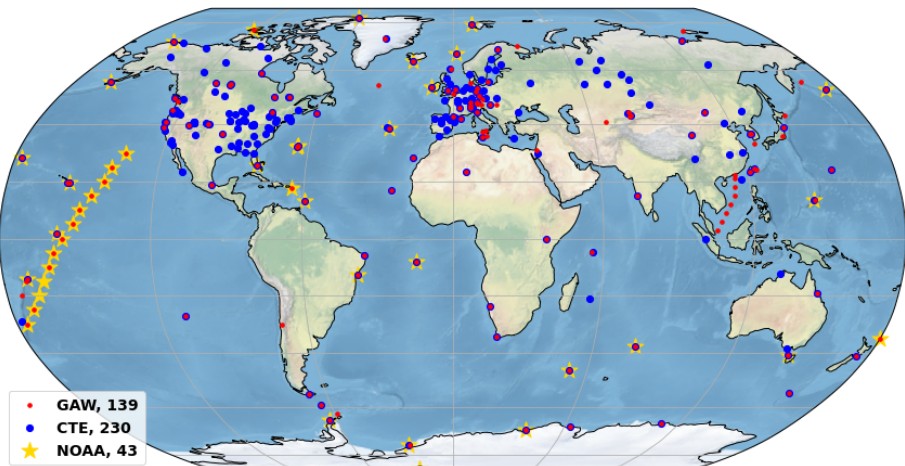

**Figure 1.** Three observation networks are employed to assess the impact of continental site inclusion when calculating global $CO_2$ mole fraction and its growth rates. The NOAA network (43 sites; yellow stars) comprises MBL sites only. The selected GAW global network (139 sites; red dots) includes both MBL sites and continental sites, for example from the Advanced Global Atmospheric Gases Experiment (AGAGE) and European ICOS contribution network. The CTE network serves as the global network for the CTE model evaluations (230 sites; blue dots), and comprises MBL sites and a more extensive inclusion of continental sites.

der a GPL license (https://doi.org/10.18160/Q788-9081, Wu, 2023). The GFIT method can be summarized and illustrated by the three steps described in the next subsections.

### 2.3.1   Fitting and filter

$CO_2$ records from each station can be abstracted as a combination of long-term trend and seasonality, which can be fitted by a function consisting of polynomial and harmonics. We applied a linear regression analysis based on three polynomial coefficients and four harmonics (Eq. 1) to fit $CO_2$ data using general linear least-squares fit (LFIT, Press et al., 1988):

$$f(x) = a_0 + a_1 t + a_2 t^2 + \ldots + a_k t^k$$
$$+ \sum_{n=1}^{n_h} (A_n \cos 2\pi n t + B_n \sin 2\pi n t), \tag{1}$$

where $a_k$, $A_n$, and $B_n$ are fitted parameters; $t$ is the time from the beginning of the observation and it is in months and expressed as a decimal of its year; $k$ denotes the polynomial number, $k = 2$; $n_h$ denotes harmonic number; and $n_h = 4$. Figure 2 illustrates the function fit to $CO_2$ data to obtain the annual oscillation (red line in Fig. 2a), is a combination of a polynomial fit to the trend (blue line in Fig. 2a), and is a harmonic fit to the seasonality (green line in Fig. 2b).

The residuals are the difference between raw data and the function fit (black dots in Fig. 2c). The filtering method is based on Thoning et al. (1989) which transforms $CO_2$ data from time domain to frequency domain using a fast Fourier transform (FFT), then applies a low pass filter to the frequency data to remove high-frequency variations, and then transforms the filtered data back to the time domain using an inverse FFT. The short-term (a cut-off value of 80 d; red line in Fig. 2c) and long-term (a cut-off value of 667 d; blue line in Fig. 2c) filters used here are the same as in NOAA method and applied to obtain the short-term and interannual variations that are not determined by the fit function. The original code is also available as Python code from the NOAA website (https://gml.noaa.gov/aftp/user/thoning/ccgcrv/; last access: 7 December 2023).

### 2.3.2   Calculate smoothed $CO_2$ and long-term trend

The result of filtering residuals is added to the fitted curve to obtain smoothed $CO_2$ and its long-term trend. The smoothed $CO_2$ comprises fitted trend, fitted seasonality and smoothed residuals (red line in Fig. 2d), the latter removes only short-term variations or noise. The long-term trend comprises fitted trend and residual trend, which removes seasonal cycle and noise (blue line in Fig. 2d).

### 2.3.3   Calculate $CO_2$ growth rate, $G_{ATM}$

The $CO_2$ growth rate ($G_{ATM}$) is determined by taking the first derivative of the long-term trend. However, the growth is made up of discrete points, e.g. the black dots in Fig. 3a show the trend points. In this case, a cubic spline interpolation is applied to the trend points, in which the spline curve passes through each trend point, such as the blue line in Fig. 3a. $G_{ATM}$ is obtained by taking the derivative of the spline at each trend point (Fig. 3b).

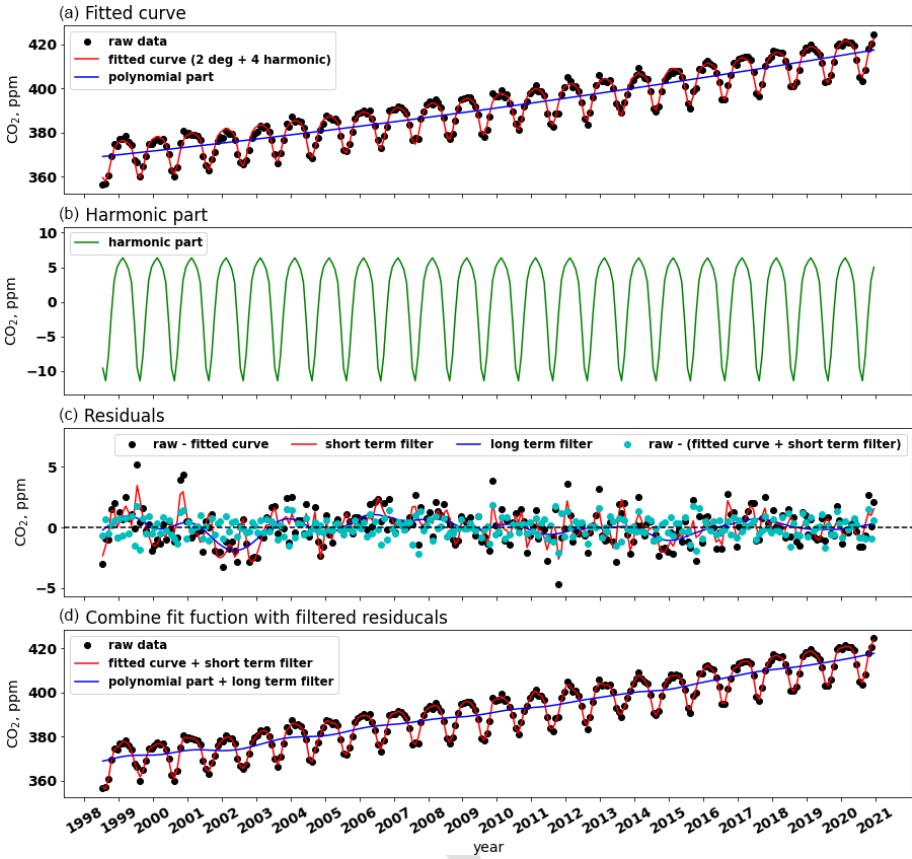

**Figure 2.** Example of analysed $CO_2$ data from station Pallas (PAL, Finland), illustrating GFIT curve fitting and filter method. Panel **(a)** shows monthly averaged $CO_2$ (dots), curve fitting with 2-degree TS4 polynomial and 4-degree harmonics (red line), and long-term trend estimated by a 2-degree polynomial (blue line). Panel **(b)** shows seasonality estimated by 4-degree harmonics. Panel **(c)** shows the residuals of raw data from the function fit (black dots). The red line is obtained by the short-term filter and the blue line is obtained by the long-term filter. The cyan dots show the residuals of raw data from the sum of fitted curve and smoothed residuals. Panel **(d)** shows final processed $CO_2$, which comprises fitted trend, fitted seasonality, and smoothed residuals (red line). The blue line shows the final trend which comprises fitted trend and residuals trend.

## 3 Results

Global averaged surface $CO_2$ and its $G_{ATM}$ are calculated using the WDCGG method and our GFIT method based on the data from the GAW and CTE networks (Fig. 1). The different observation networks and their analysis methods are listed in Table 1. We calculated the global means and its $G_{ATM}$ by area-weighted averaging the zonal means over each latitudinal band (30°), following the same $CO_2$ averaging method as described in Tsutsumi et al. (2009). A bootstrap method is used to estimate the uncertainties of global $CO_2$ mean and its $G_{ATM}$, which is an almost identical uncertainty analysis as presented by Conway et al. (1994) who constructed 100 bootstrap networks for the NOAA analysis. We construct 200 bootstrap networks, consistent with the WDCGG analysis in Tsutsumi et al. (2009). For each bootstrap network, we randomly draw the same number of sites as the actual network (e.g. 139 sites for GAW network) with replacement from the actual network, which means some sites are miss-

ing whereas others are represented twice or more often. We calculate global mean $CO_2$ mole fraction and its $G_{ATM}$ for each network and then calculate the statistics (i.e. mean and 68 % confidence interval (CI)) on the 200 networks. All uncertainties in this paper are reported as $\pm 68$ % CI.

### 3.1 Globally averaged surface $CO_2$ mole fraction and its $G_{ATM}$

Figure 4 presents a monthly comparison of globally and locally averaged $CO_2$ mole fractions and their $G_{ATM}$ from 1980 to 2020. The statistical metrics assessing the agreement of these monthly comparisons are available in Fig. 5 (for 2001–2020) and Fig. S1 in the Supplement (for 1980–2020). The statistical metrics for the annual comparisons can be found in Fig. S2 (for 2001–2020) and Fig. S3 (for 1980–2020). They exhibit a similar pattern to the monthly comparisons (i.e. Figs. 5 and S1).

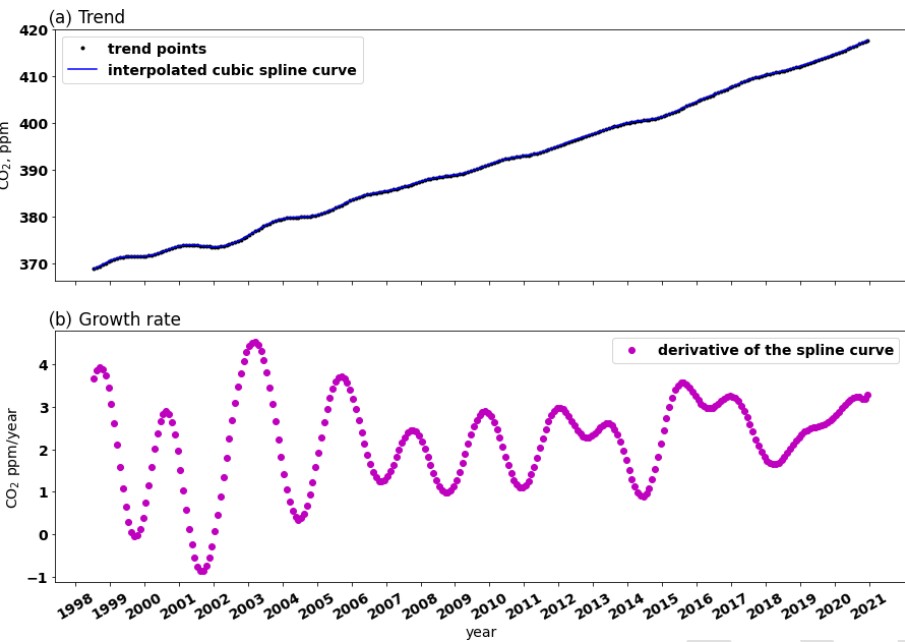

**Figure 3.** Example of $CO_2$ growth rate. The raw data is the same as in Fig. 2 from station Pallas (PAL, Finland). Panel **(a)** shows the trend points (black dots) CE1 and its cubic spline interpolation (blue line). Panel **(b)** shows the $G_{ATM}$ at each trend point.

**Table 1.** Description of the three observation networks and their analysis methods.

| Terminology | Description |
|---|---|
| NOAA network | NOAA network comprises MBL sites only (43 sites). |
| GAW network | The selected GAW global network (139 sites) includes both MBL sites and some continental sites. |
| CTE network | The CTE network serves as the global network for the CTE model evaluations (230 sites), and comprises MBL sites and a more extensive inclusion of continental sites. |
| GAW (GFIT) | GAW network observations analysed using the GFIT method. |
| GAW (WDCGG) | GAW network observations analysed using the WDCGG method without extrapolation. |
| GAW (WDCGG+) | GAW network observations analysed using the WDCGG method with extrapolation. |
| CTE_obs (GFIT) | CTE network observations analysed using the GFIT method. The observations come from the ObsPack data product (Schuldt et al., 2022). |
| CTE_output (GFIT) | CTE model output at the 230 sites (sampled at the same location, altitude, and time) analysed using the GFIT method. |
| CTE_global (GFIT) | CTE model output for full global grids (averaged over the first three levels, 0–0.35 km altitude) analysed using the GFIT method. |
| MLO (GFIT) | Mauna Loa (MLO) observations analysed using the GFIT method. |
| SPO (GFIT) | South Pole (SPO) observations analysed using the GFIT method. |

Globally averaged monthly surface $CO_2$ mole fractions, derived from the GAW network (GAW (GFIT) or GAW (WDCGG)), are significantly ($p < 0.05$) higher by 0.329–0.335 ppm during 1980–2020 (Fig. S1a) and 0.370–0.390 ppm during 2001–2020 (Fig. 5a) when compared with the NOAA analysis (Fig. 4a). This finding aligns with that of Tsutsumi et al. (2009), who reported a 0.350 ppm higher global average in the GAW network during 1983–2006. The higher estimate from the GAW network can be attributed to the inclusion of more diverse sites, encompassing not only NOAA's MBL sites but also additional continental sites (Fig. 1).

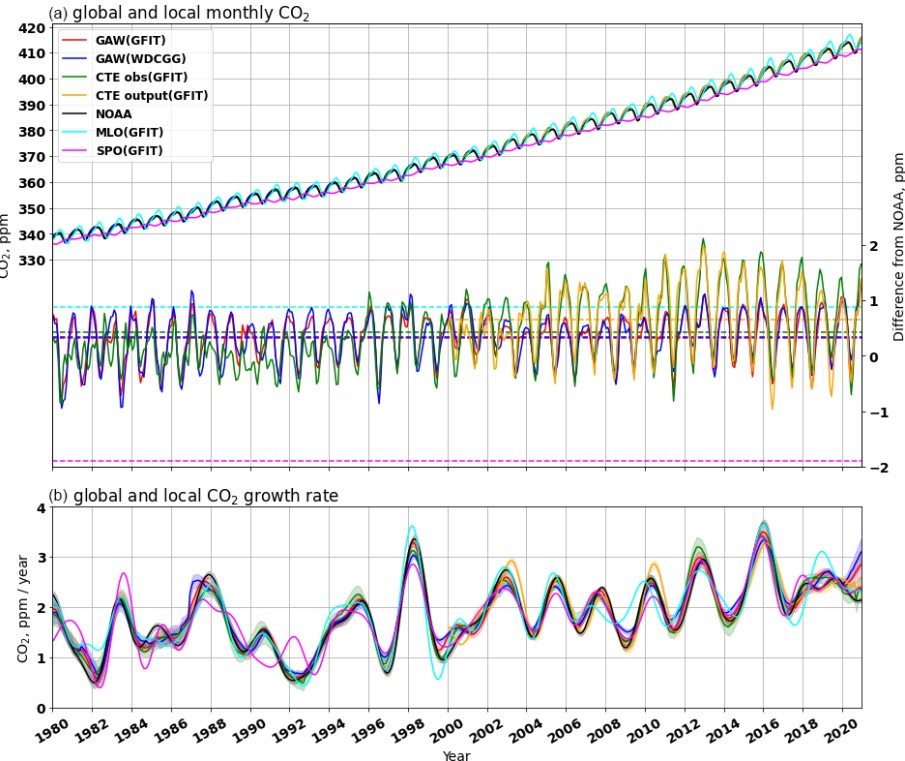

**Figure 4.** Comparison of globally and locally averaged CO$_2$ mole fraction **(a)** and its $G_{ATM}$ **(b)** from 1980 to 2020. Panel **(a)** shows the global monthly CO$_2$ mole fraction from 139 GAW sites (estimated from observations only), 43 NOAA MBL sites, and those from 230 sites used in CTE (either from observations or model output). The two local CO$_2$ mole fractions are from Mauna Loa (MLO; cyan line) and South Pole (SPO; magenta line) stations, analysed using the GFIT method. The red and blue lines show the CO$_2$ derived from GAW (GFIT) and GAW (WDCGG), respectively. The green and orange lines show the CO$_2$ derived from CTE_obs (GFIT) and CTE_output (GFIT), respectively. The right $y$ axis shows their difference from NOAA CO$_2$ mole fraction, and the dashed lines show the mean of the difference over the available period. Panel **(b)** compares the corresponding global and local CO$_2$ growth rate; the legend refers to **(a)**. The shadow area shows the uncertainty as a 68 % confidence interval obtained by the bootstrap analysis.

Both global CO$_2$ and its $G_{ATM}$ derived from the GAW (GFIT) and GAW (WDCGG) are nearly overlapping (the red and blue lines) in Fig. 4a and b. The statistical metrics (Figs. 5 and S1) indicate a high agreement (ME < 0.020 ppm, RMSE < 0.145 ppm, and $r$ > 0.999 for CO$_2$ mole fraction; ME < 0.005 ppm yr$^{-1}$, RMSE < 0.108 ppm yr$^{-1}$, and $r$ > 0.982 for $G_{ATM}$) between these two methods, which confirms that the GFIT method agrees well with WDCGG method without extrapolation. The WDCGG method with extrapolation (i.e. GAW (WDCGG+)), which involves extrapolating the long-term trend of each station to match the period of the most long-running station and adding it to the average seasonal variation to synchronize data period of all stations (Tsutsumi et al., 2009), produces 0.096 ppm significantly ($p$ < 0.05) higher values than the global monthly surface CO$_2$ mole fraction derived from the GAW (WDCGG) during the common period 1984–2020 (Table S1 in the Supplement). However, the extrapolation has a minimal effect (RMSE = 0.076 ppm yr$^{-1}$ and ME = −0.011 ppm yr$^{-1}$; Table S1) on the CO$_2$ growth rate.

Globally averaged monthly surface CO$_2$ derived from CTE_obs (GFIT) and CTE_output (GFIT) are 0.422 ppm (1980–2020; Fig. S1) and 0.668 ppm (2001–2020; Fig. 5) significantly ($p$ < 0.05) higher compared with the NOAA analysis, respectively (Fig. 4a). Comparing the global mean of CTE_obs (GFIT) with CTE_output (GFIT) during the common period of 2001–2020, we observe a low bias (0.069 ppm in CTE_output; Fig. 5a), which suggests that the CTE model results can reasonably reproduce the global mean CO$_2$ levels. The global annual CO$_2$ mole fraction from CTE_obs (GFIT), CTE_output (GFIT), and CTE_global (GFIT) is 0.367, 0.299, and 0.186 ppm significantly ($p$ < 0.05) higher than the result of the GAW (GFIT), respectively (Fig. 5a). The higher global mean from CTE_obs (GFIT) and CTE_output (GFIT) can be attributed to the presence of more sites in the Northern Hemisphere within the CTE network compared with the GAW network. The lower bias observed between GAW (GFIT) and CTE_global (GFIT) suggests that the GAW network provides a good representation of the low-level atmosphere (i.e. 0 to 0.35 km altitude) at global scale, or the CTE model performs well in the low-level atmosphere.

A common approach to estimate global surface $CO_2$ mole fraction is by using one or two representative sites, such as MLO and SPO. The globally averaged monthly surface $CO_2$ mole fractions, derived from the GAW, CTE, and NOAA networks, are significantly ($p < 0.05$) lower by 0.46–0.88 ppm during 1980–2020 (Fig. S1a) and 0.45–1.19 during 2001–2020 (Fig. 5a) than the local $CO_2$ estimates solely based on MLO measurements. Conversely, these global monthly $CO_2$ mole fractions are significantly ($p < 0.05$) higher by 1.91–2.24 ppm during 1980–2020 (Fig. S1a) and 2.21–2.94 ppm during 2001–2020 (Fig. 5a) when compared with local measurements at the SPO site. Furthermore, the global seasonal cycle leads the local cycle at MLO by approximately 1 month (estimated by averaging the time difference between the peaks of their seasonal cycles). In contrast, the local cycle at SPO is not evident and is opposite to the global seasonal cycle (Fig. 4a).

Despite differences in the global averaged surface $CO_2$ mole fractions derived from different networks and analysis methods, the $G_{ATM}$ derived from GAW network, CTE network and its model output, and NOAA network exhibits strong agreement during 1980–2020 (ME $< 0.031$ ppm yr$^{-1}$, RMSE $< 0.217$ ppm yr$^{-1}$, and $r > 0.948$; Figs. 4b and S1). The differences in the $G_{ATM}$ remain below 0.023 ppm yr$^{-1}$ during 2001–2020, with low or no significance level (Fig. 5b), especially when comparing the annual $G_{ATM}$ (Fig. S2b). Furthermore, over the long-term period of 40 years, the estimated local growth rate at MLO (ME $< 0.046$ ppm yr$^{-1}$ higher, RMSE $< 0.272$ ppm yr$^{-1}$, and $r > 0.915$) and SPO (ME $< 0.049$ ppm yr$^{-1}$ lower, RMSE $< 0.305$ ppm yr$^{-1}$, and $r > 0.888$) behaves similarly to the $G_{ATM}$ derived from the GAW, CTE, and NOAA networks (Figs. 4b and S1). However, noticeable monthly differences between the local and global growth rates, deviating up to approximately 0.8 ppm yr$^{-1}$, and time shifts are observed (Fig. 4b).

The trend analysis reveals that with development of continental sites, the slope of the trend of annual global $CO_2$ mole fraction changes from the NOAA network ($1.832 \pm 0.029$ ppm yr$^{-1}$) to the CTE network ($1.859 \pm 0.029$ ppm yr$^{-1}$) during 1980–2020 (Fig. S4). However, the $G_{ATM}$ increased steadily at a rate of $0.030 \pm 0.002$ ppm yr$^{-1}$ each year from 1980 to 2020 (Fig. 6a), based on the observations from the three networks (i.e. GAW, CTE, and NOAA). This implies that over long-term periods (here 40 years), the networks with and without continental sites exhibit the same trend in the $G_{ATM}$ and have little effect on the transient change in the rate of $CO_2$ increase in the atmosphere. Hence, the role of $CO_2$ advective transport and mixing in estimating the long-term change in the $G_{ATM}$ appears negligible. However, a notable difference emerges in the short-term (here 1 month) change in the $G_{ATM}$ between the networks with and without continental sites (Fig. 6b). El Niño events are known to diminish net global C uptake (due to factors such as droughts, floods, and fires) while increasing global $CO_2$ growth rate (Sarmiento et al., 2010). During three strong El Niño events, marked as grey bands in Fig. 6b, the $G_{ATM}$ derived from the GAW and CTE networks (red and green lines) begins to increase before the El Niño events (marked as blue circles in Fig. 6b), approximately 1–2 months earlier than that derived from the NOAA network (black line) and it also reaches its peak during El Niño events (marked as orange circles in Fig. 6b) about 1–2 months earlier (Table S2). CE2 This suggests that continental sites can aid in the early detection of $G_{ATM}$ changes resulting from changes in biogenic emission or uptake. The CTE network (green line) even detects the change 1 month earlier than the GAW network (red line), e.g. for the El Niño 1997–1998 event (Fig. 6b; Table S2). This earlier detection is attributed to the inclusion of even more continental sites in the CTE network (Fig. 1), although the more continental sites also induce the greater variability.

Table 2 presents the global annual $CO_2$ mole fraction and its $G_{ATM}$ derived from GAW (GFIT), along with the uncertainty estimates using the bootstrap method. The global average surface $CO_2$ mole fraction increased from TS5 $339.17 \pm 0.38$ ppm in 1980 to $413.06 \pm 0.16$ ppm in 2020. Notably, the uncertainty is greater before 1990, primarily due to the limited number of measurement stations worldwide during that period. The average $G_{ATM}$ for the two decades before 2000 is approximately $1.54 \pm 0.08$ ppm yr$^{-1}$. However, in the subsequent two decades, has experienced increases, reaching $1.91 \pm 0.05$ ppm yr$^{-1}$ during 2000–2009 and further rising to $2.41 \pm 0.06$ ppm yr$^{-1}$ during 2010–2019 (Fig. S5; Table 2).

## 3.2   Vertical profile of global $CO_2$ mole fraction

The CTE model simulates $CO_2$ mole fraction on global 3D grids, enabling us to visualize the modelled vertical $CO_2$ profile. In the lower atmosphere, highest $CO_2$ mole fraction is found in the northern mid-latitude region (dark red between 30 and 40° N; Fig. 7a). This area experiences more anthropogenic emissions, which are subsequently transported towards both northern and southern latitudes. The latitudinal and interhemispheric gradient of atmospheric $CO_2$, as shown in Fig. 7a, is influenced not only by differences in the latitudinal and interhemispheric fossil fuel emissions and seasonal exchanges with terrestrial biota (Denning et al., 1995), but also by atmospheric transport (Patra et al., 2011). As altitude increases, the gradient between the Northern and Southern hemispheres becomes small and levels out at higher altitudes (e.g. $> 50$ km). When comparing the vertical profile change between 2001 and 2020 (Fig. 7b and c), we observe that the $CO_2$ mole fraction increases slowly in the higher atmosphere ($> 25$ km altitude) compared with the lower atmosphere ($< 25$ km altitude). Figure 7c shows that the vertical gradient (difference between 50 and 0.05 km) changes from approximately 5 ppm in 2001 to around 13 ppm in 2020. The high vertical gradient in 2020 reflects the accumulation of

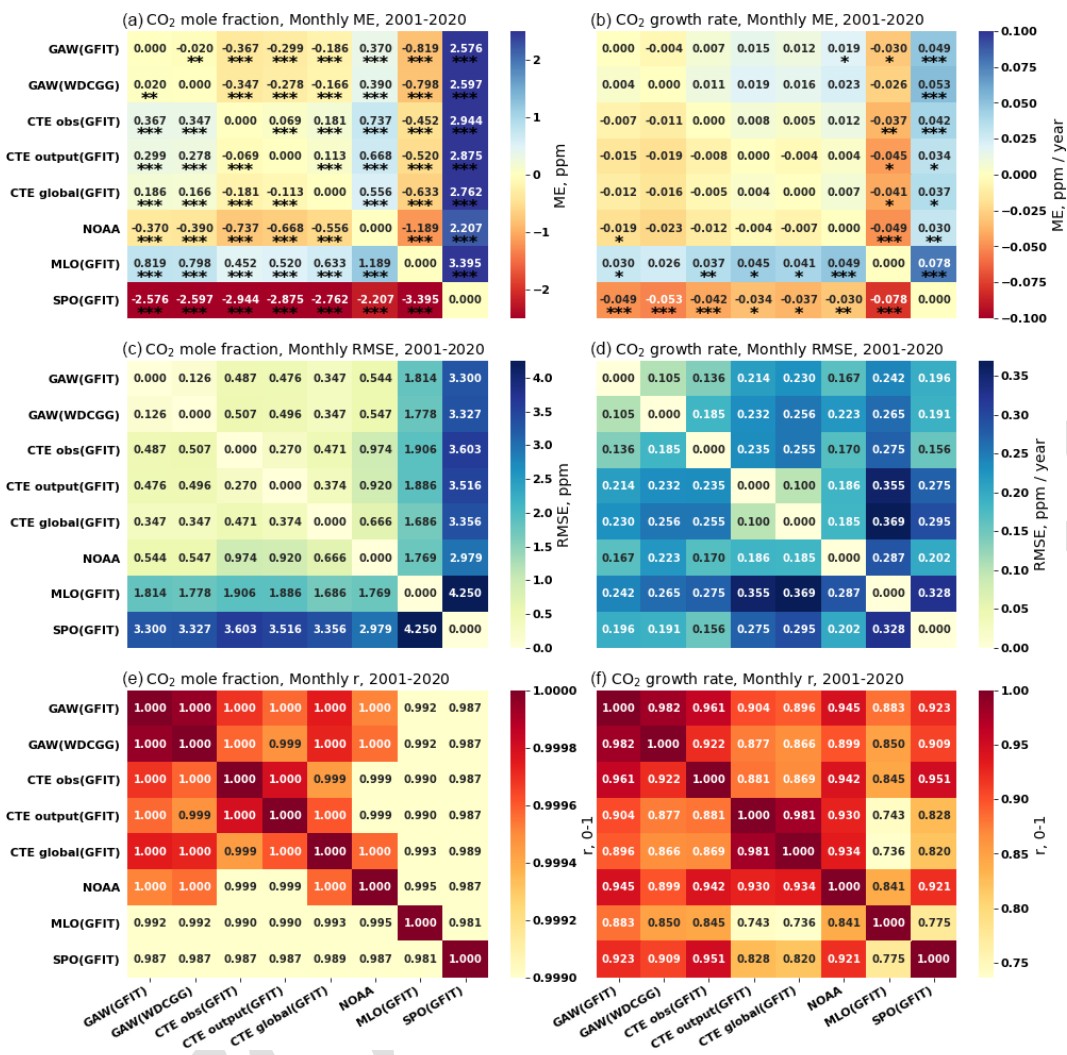

**Figure 5.** Pair-wise statistical metrics assess the agreement of monthly global and local $CO_2$ mole fraction (ppm) and its $G_{ATM}$ (ppm yr$^{-1}$) across various networks and methodologies (see Fig. 4 and Table 1) for the period 2001–2020. Panel **(a)** presents the mean error (ME) quantifying the difference for each pair, focusing on $CO_2$ mole fraction, while **(b)** does the same for $G_{ATM}$. The significance levels of paired $t$ test for ME are indicated as follows: * $p < 0.1$, ** $p < 0.05$ and *** $p < 0.01$. Panels **(c)** and **(d)** present the root mean squared error (RMSE) for $CO_2$ mole fraction and $G_{ATM}$, respectively. Panels **(e)** and **(f)** present the Pearson correlation coefficient ($r$) for $CO_2$ mole fraction and $G_{ATM}$, respectively.

$CO_2$ in the lower atmosphere, resulting from continuous $CO_2$ emissions from the surface during 2001–2020 and slow vertical transport. The low vertical gradient in 2001 is partly due to lower surface emissions.

Pressure-weighted average $CO_2$ mole fraction in the lower atmosphere (0–0.35 km altitude) and the entire atmosphere are calculated from CTE output. The annual absolute change in $CO_2$ mole fraction, computed as the difference between annual means, is more pronounced in the lower atmosphere (orange bars in Fig. S6a) than in the entire atmosphere (blue bars in Fig. S6a). The reason is that the entire atmosphere has a larger air volume than the lower atmosphere, and changes in the surface $CO_2$ sinks and sources are diluted due to atmospheric horizontal and vertical transport. The $CO_2$ annual

absolute change derived from GAW (GFIT), GAW (WDCGG), and NOAA (represented by red, purple, and brown bars in Fig. S6a) shows small positive or negative differences from the CTE_output (GFIT) and CTE_global (GFIT) across different years. However, over the long term (e.g. on a decadal scale, 2001–2010 and 2011–2020), the CTE model-derived changes in lower and entire atmospheric $CO_2$ show good agreement ($< 0.09$ ppm yr$^{-1}$) with the surface observation-based estimate, especially for lower atmospheric $CO_2$ ($< 0.07$ ppm yr$^{-1}$). In Fig. S6b, the interannual variability (IAV) of $CO_2$ mole fraction derived from the CTE model follows a similar temporal pattern as the observation-based IAV derived from the GAW and NOAA network, and especially the IAV of the low-level atmosphere (orange bars)

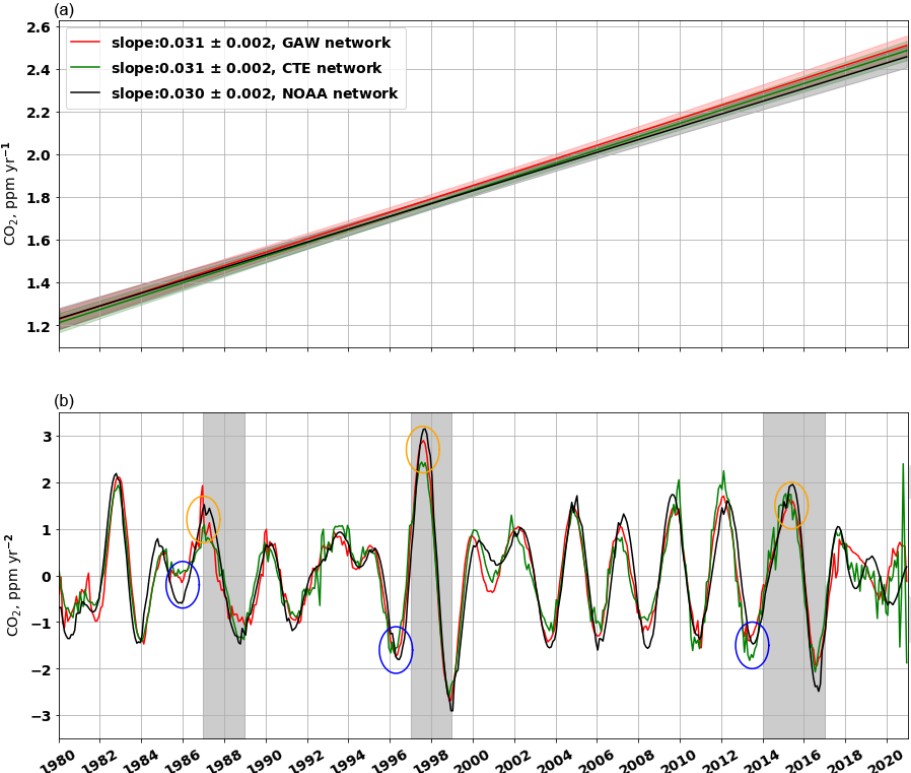

**Figure 6.** Trend analysis of the global $CO_2$ growth rate from 1980 to 2020. Panel **(a)** shows the trends in $CO_2$ growth rate for the GAW network (red line), the CTE network (green line), and the NOAA network (black line) during the whole period 1980–2020. The $CO_2$ growth rate is derived from GAW (GFIT), CTE_obs (GFIT), and NOAA analysis (Fig. 4b). Panel **(b)** shows the trend in $CO_2$ growth rate for each month during 1980–2020, calculated as the derivative of the growth rate. The grey bands mark the period of three strong El Niño events, i.e 1987–1988, 1997–1998, and 2014–2016.

exhibits strong agreement with the observation-based IAV ($r > 0.971$ and RMSE $< 0.178$ ppm).

### 3.3 Relationship between the surface $CO_2$ mole fraction and atmospheric $CO_2$ mass

The atmospheric $CO_2$ mass, calculated from the CTE output as a function of air mass and $CO_2$ concentration (Sect. S3), has increased from 789.46 PgC in 2001 to 877.88 PgC in 2020 (Fig. S7a). The spatial distribution of the atmospheric $CO_2$ mass is presented in Fig. S7b and c. Monthly global surface $CO_2$ mole fraction derived from CTE_output (GFIT) and GAW (GFIT), represented as red and blue dots in Fig. 8a, exhibit a similar linear relationship with the monthly atmospheric $CO_2$ total mass, both showing the same slope of $2.08 \pm 0.01$ PgC ppm$^{-1}$. Similarly, NOAA $CO_2$ (green dots in Fig. 8a) also demonstrates a comparable linear relationship with a slope of $2.09 \pm 0.01$ PgC ppm$^{-1}$. Notably, the slopes or conversion factors in Fig. 8a are slightly lower than the factor 2.12 PgC ppm$^{-1}$ used in Ballantyne et al. (2012) for the period 1980–2010. This minor difference in the conversion factor is expected, considering the different model and data used.

We further compare the interannual variability (IAV), calculated as the anomaly departure from a quadratic trend, of the atmospheric $CO_2$ mass and the surface $CO_2$ (Fig. 8b). The coefficient of the linear relationship closely approaches $\sim 1.0$, indicating that the temporal changes in atmospheric $CO_2$ mass align with the temporal changes in surface $CO_2$ mole fraction. The $CO_2$ IAV based on the NOAA network exhibits a slightly closer relationship ($r = 0.938$) with the CTE atmospheric $CO_2$ mass estimates than the GAW ($r = 0.861$) and CTE ($r = 0.812$) networks. This finding is consistent with the long atmospheric residence time and well-mixed nature of $CO_2$ in the NOAA network. Overall, the relationship found in Fig. 8 implies that the current surface $CO_2$ network can effectively serve as an indicator of the $CO_2$ mass changes throughout the entire atmosphere through a linear relationship.

## 4 Discussion

Over the past few decades, observational networks have been extended beyond the NOAA MBL network to include more continental sites, such as in the GAW and CTE networks (Fig. 1). These expansions aim to better monitor global $CO_2$

**Table 2.** Annual[TS6] global averaged $CO_2$ mole fraction (mean in ppm) and its $G_{ATM}$ (in ppm yr$^{-1}$) derived from GAW observations using the GFIT method. $U$(Mean) and $U$($G_{ATM}$) respectively indicate the uncertainty of the mean and its $G_{ATM}$ as a 68 % confidence interval. The annual value is averaged over the monthly values of the year.

| Year | 1980 | 1981 | 1982 | 1983 | 1984 | 1985 | 1986 | 1987 | 1988 | 1989 |
|---|---|---|---|---|---|---|---|---|---|---|
| Mean | 339.17 | 340.16 | 341.03 | 342.59 | 344.46 | 345.69 | 347.08 | 348.99 | 351.45 | 353.15 |
| $U$ (Mean) | 0.38 | 0.24 | 0.19 | 0.24 | 0.26 | 0.22 | 0.14 | 0.15 | 0.12 | 0.15 |
| $G_{ATM}$ | 1.65 | 1.07 | 0.88 | 2.02 | 1.32 | 1.38 | 1.55 | 2.38 | 2.08 | 1.23 |
| $U$ ($G_{ATM}$) | 0.12 | 0.10 | 0.15 | 0.13 | 0.08 | 0.11 | 0.14 | 0.08 | 0.09 | 0.06 |
| Year | 1990 | 1991 | 1992 | 1993 | 1994 | 1995 | 1996 | 1997 | 1998 | 1999 |
| Mean | 354.22 | 355.64 | 356.37 | 357.09 | 358.51 | 360.52 | 362.27 | 363.40 | 366.14 | 368.10 |
| $U$ (Mean) | 0.10 | 0.11 | 0.10 | 0.10 | 0.11 | 0.12 | 0.12 | 0.10 | 0.10 | 0.10 |
| $G_{ATM}$ | 1.41 | 1.03 | 0.65 | 1.22 | 1.72 | 2.06 | 1.16 | 1.82 | 2.89 | 1.34 |
| $U$ ($G_{ATM}$) | 0.08 | 0.06 | 0.05 | 0.05 | 0.05 | 0.08 | 0.07 | 0.05 | 0.05 | 0.05 |
| Year | 2000 | 2001 | 2002 | 2003 | 2004 | 2005 | 2006 | 2007 | 2008 | 2009 |
| Mean | 369.30 | 370.77 | 372.92 | 375.45 | 377.22 | 379.28 | 381.38 | 383.20 | 385.26 | 386.78 |
| $U$ (Mean) | 0.12 | 0.11 | 0.10 | 0.10 | 0.10 | 0.10 | 0.09 | 0.10 | 0.10 | 0.11 |
| $G_{ATM}$ | 1.58 | 1.58 | 2.33 | 2.17 | 1.66 | 2.42 | 1.75 | 2.20 | 1.71 | 1.68 |
| $U$ ($G_{ATM}$) | 0.05 | 0.06 | 0.06 | 0.04 | 0.04 | 0.03 | 0.05 | 0.04 | 0.05 | 0.04 |
| Year | 2010 | 2011 | 2012 | 2013 | 2014 | 2015 | 2016 | 2017 | 2018 | 2019 |
| Mean | 389.01 | 390.97 | 393.14 | 396.00 | 397.79 | 400.12 | 403.47 | 405.70 | 407.93 | 410.57 |
| $U$ (Mean) | 0.12 | 0.12 | 0.14 | 0.11 | 0.10 | 0.10 | 0.11 | 0.09 | 0.10 | 0.13 |
| $G_{ATM}$ | 2.32 | 1.73 | 2.74 | 2.30 | 1.91 | 2.98 | 2.95 | 2.04 | 2.50 | 2.61 |
| $U$ ($G_{ATM}$) | 0.05 | 0.06 | 0.09 | 0.05 | 0.04 | 0.05 | 0.06 | 0.06 | 0.07 | 0.05 |
| Year | 2020 | | | | | | | | | |
| Mean | 413.06 | | | | | | | | | |
| $U$ (Mean) | 0.16 | | | | | | | | | |
| $G_{ATM}$ | 2.60 | | | | | | | | | |
| $U$ ($G_{ATM}$) | 0.16 | | | | | | | | | |

concentrations and quantify $CO_2$ sources and sinks. While the continental observations encompass contributions from both substantial sources of anthropogenic emissions and sources/sinks from terrestrial vegetation and soil, these continental observations consistently yield a higher global surface $CO_2$ mole fraction in the overall global $CO_2$ analysis, indicating that they are influenced by a bigger net source. We find that the global mean derived from the GAW network is consistently 0.329 (GFIT method) or 0.335 (WD-CGG method) ppm higher than that derived from the NOAA network during 1980–2020. Similarly, Tsutsumi et al. (2009) reported a roughly 0.350 ppm higher mole fraction in the GAW network for the years 1983–2006. Notably, the CTE network leads to an even higher global mean (0.422 ppm during 1980–2020), which is likely due to more observational sites located in the Northern Hemisphere, where the highest anthropogenic emissions occur. This also explains the large fluctuation of $CO_2$ concentrations observed during the winters and summers during 2001–2020 (Fig. 4a). In the future, with the addition of new observation sites, particularly in the Northern Hemisphere, to the existing observational net-

work (e.g. GAW network), we expect that this would lead to higher global surface $CO_2$ levels and a greater amplitude in the global $CO_2$ seasonal cycle in the global $CO_2$ analysis.

Although Friedlingstein et al. (2022) reported a 5.4 % drop ($\sim 0.52$ PgC) in fossil fuel $CO_2$ emissions in 2020 (due to restrictions on transport, industry, power, etc., during the COVID-19 pandemic), the increase in annual $CO_2$ from 2019 to 2020 ($2.60 \pm 0.16$ ppm yr$^{-1}$) remains at a similar level as from 2018 to 2019 ($2.61 \pm 0.05$ ppm yr$^{-1}$). In principle, an equivalent drop of roughly 0.25 ppm yr$^{-1}$ (according to the conversion factor 2.08 PgC ppm$^{-1}$ in Fig. 8a) or roughly 0.13 ppm yr$^{-1}$ (according to the annual absolute change; red bars in Fig. S6a) in the growth rate should be visible for the period 2019–2020 due to the declined $CO_2$ emissions. However, such a short-term human activity induced change in the $CO_2$ growth rate may be hidden by the natural variability. The bootstrap analysis is used in this study (also in Conway et al., 1994, and Tsutsumi et al., 2009) to estimate the uncertainty of the $CO_2$ temporal mean and its growth rate and to assess how sensitive the global value is to the distribution of sampling sites. The relatively large uncertainty

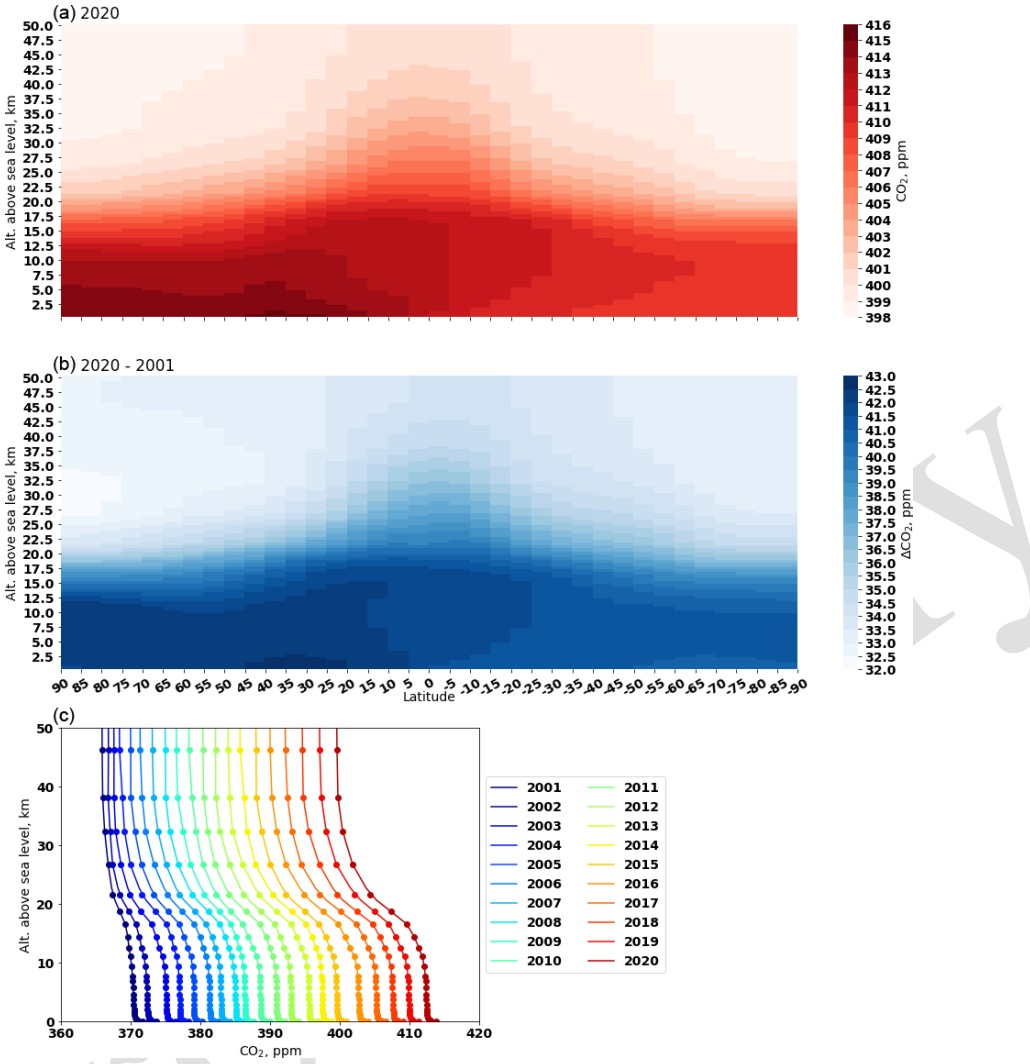

**Figure 7.** Global vertical profile of $CO_2$ mole fraction derived from CTE model output. Panel (**a**) presents the vertical profile in 2020. Panel (**b**) presents the difference in the vertical profile between 2001 and 2020. Panel (**c**) presents the annual mean vertical profile from 2001 to 2020. The dots mark CTE vertical level heights and the lines are the linear interpolation between the heights.

($\pm 0.16\,\mathrm{ppm}\,\mathrm{yr}^{-1}$) at the end of 2020 compared with previous years (Table 2) is likely due to an end-effect associated with the curve fitting and filter procedure. The end-effect is a tendency for the growth rate to converge toward the mean value at the end of the record (Conway et al., 1994). Therefore, Conway et al. (1994) suggested that the growth rate curves for the last 6 months should be viewed with caution. Reducing the end effect requires further study, such as using machine learning or bias-correction methods to extrapolate the smoothed trend for a short period (e.g. 1 year) before and after. This extrapolated portion is used exclusively for calculating local mole fraction and growth rate, while it is not included in the global or zonal average, as it could introduce additional uncertainty.

Extrapolation beyond the measurement period extends knowledge gained from a limited period of measurements.

During a limited measurement period, we can define the average seasonality, long-term trend, and short-term variation at a measurement site. The long-term trend of an individual site can be extrapolated by various methods, such as referring to the latitude reference time series (Masarie and Tans, 1995) or calculating the mean long-term trend over sites within a certain latitudinal zone (e.g. 30°) (Tsutsumi et al., 2009). This extrapolated trend is then combined with the average seasonality to produce estimates beyond the measurement period. However, the extrapolation process relies on the assumption that the relationship of an individual site to the latitude reference remains invariant in time, while in reality the relationship between nearby sites is continuously changing (Masarie and Tans, 1995). In addition, the short-term variation is often ignored or estimated from nearby sites, introducing extra uncertainty into the extrapolation process. In

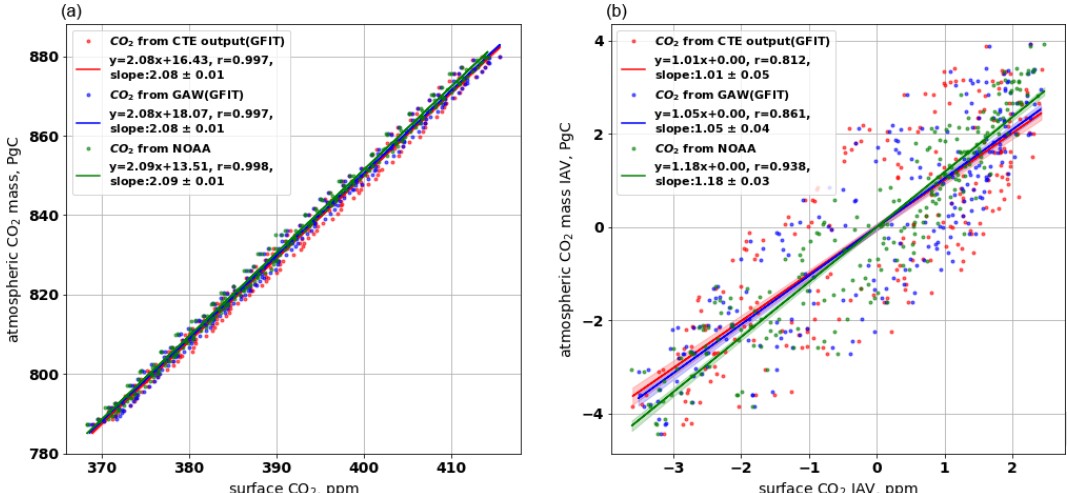

**Figure 8.** Relationship between the monthly surface CO$_2$ mole fraction and atmospheric CO$_2$ mass. The atmospheric CO$_2$ mass calculated from the 3D CTE output. In **(a)**, the monthly surface CO$_2$ is derived from the CTE_output (GFIT), GAW (GFIT), and NOAA analysis, presented as blue, red, and green dots, respectively. Panel **(b)** compares the corresponding interannual variability (IAV) of the atmospheric CO$_2$ mass and the surface CO$_2$. The IAV is calculated as the anomaly departure from a quadratic trend.

this study, we find that the WDCGG method with extrapolation (GAW (WDCGG+)) results in a global surface CO$_2$ mole fraction approximately 0.096 ppm higher than the WDCGG method without extrapolation (GAW (WDCGG)) using the same GAW observations, although the extrapolation has a minor effect on the growth rate (Table S1). Therefore, we chose not to use extrapolation beyond the measurement period in our analysis. As the number of long-term measurements increases, the need for such extrapolation becomes less necessary.

Our analysis shows that basing the CO$_2$ growth rate on GAW surface observations does not introduce a large bias (with an average agreement within 0.016 ppm yr$^{-1}$) compared with a full atmospheric analysis (Figs. 4b and 5). This full atmosphere CO$_2$ was provided by the CTE model, in which the global annual mean CO$_2$ is significantly overestimated compared with GAW observations (e.g. 0.299 ppm higher in CTE_output (GFIT), or 0.186 ppm higher in the CTE_global (GFIT) during 2001–2020). The overestimate derived from the CTE_output (GFIT) is mainly due to more sites in the Northern Hemisphere in the CTE network than in the GAW network. The lower overestimate derived from the CTE_global (GFIT) implies that the biases in CTE outputs are not uniform spatially and tend to balance out. We estimate the CTE bias by comparing the observations and CTE outputs at the same sites, which results in a 0.069 ppm low bias derived from the CTE outputs in calculating the global surface CO$_2$ mole fraction.

The local growth rate at MLO and SPO generally behaves similarly to the global growth rate derived from the GAW, CTE, and NOAA networks (Figs. 4b and S1). However, the local CO$_2$ mole fraction and its seasonal cycle noticeably differ from global estimates derived from different observa-

tional networks. In this regard, the utilization of individual sites for the evaluation of the global average mole fraction and its growth rate is not precise and can only be used for illustration rather than as a substitute for the proper global average calculation. The local observation sites, often situated away from significant local sources and sinks, such as MLO, provide long-term and high-quality data, serving as reference data for the global CO$_2$ mole fraction. However, a single observation site cannot capture the CO$_2$ spatial variability, transport, and mixing. To overcome these limitations, global CO$_2$ trends and variations are best assessed by integrating data from multiple sources and locations.

Different observational networks (i.e. NOAA, GAW, and CTE) are analysed in this study, revealing differences in calculated global surface CO$_2$ mole fractions equivalent to the current atmospheric growth rate over a 3-month period. This suggests that the station selection, especially if and how many continental observations are used, has some influence on the derived global surface CO$_2$ levels, but it is not particularly strong. Nowadays, an increasing number of continental observations are established to monitor biogenic sources and sinks, providing further insight into the climate change and the associated ecosystem processes (Ciais et al., 2005; Ramonet et al., 2020). Such continental observations carry more variability in measurements than the marine observations, which require caution when including them in the mix of stations used to determine global surface CO$_2$ mole fraction. Our study demonstrates that continental sites can help in early detection of changes in CO$_2$ growth rate caused by biogenic emission change, such as those resulting from El Niño events. Furthermore, the current observational networks (with and without continental sites) and CTE model show a good agreement on the global CO$_2$ growth rate, with

low or no significant differences within 0.023 ppm yr$^{-1}$ during 2001–2020 and 0.031 ppm yr$^{-1}$ during 1980–2020. This implies that the current observation networks (as shown in Fig. 1, representing various ecosystems, sinks, sources, and latitudes) have a similar good capacity to capture changes in the global surface $CO_2$, although there is the spatial and temporal variability in the $CO_2$ growth rate (e.g. Conway et al., 1994).

We also notice that the uncertainty in global $CO_2$ growth rate is approximately 0.07 ppm yr$^{-1}$, as derived from GAW (GFIT) and averaged over 1980–2020 (Table 2). To reduce the uncertainty to 0.02 ppm yr$^{-1}$ (equivalent to 1 % of the global $CO_2$ growth rate), in principle it would theoretically require adding more stations to the current observation network. We conducted an experiment that demonstrates how the uncertainty of the global $CO_2$ growth rate exponentially increases as the number of land observation sites decreased (Fig. S8). According to our experiment, to achieve the goal of reducing the uncertainty to 0.02 ppm yr$^{-1}$, 332 land observation sites are required (Fig. S8). However, the required number of sites also depends on their measurement accuracy, consistency, and geographical distribution (i.e. the $CO_2$ footprint coverage of the observation network and the importance of the network design have been addressed by Storm et al., 2023).

## 5   Conclusions

The WMO GAW $CO_2$ network documents the gradual global accumulation of $CO_2$ in the atmosphere due to human activities. It has been used to assess the large-scale and long-term environmental consequence of fossil $CO_2$ emission and land use changes. The high-quality observations conducted by the WMO GAW network include not only background stations (most of the NOAA MBL stations) but also continental stations. This comprehensive network enables proper global average calculation. Furthermore, the WMO has initiated a new programme, Global Greenhouse Gas Watch (GGGW), with the aim of establishing a reference network. This network will be built on the high-quality observations already performed under the WMO GAW programme that follows consistent good practices and standards. Although the current monitoring networks have limitations in terms of geographical coverage, data consistency, and long-term measurements, they are well equipped and have the capacity to effectively represent global surface $CO_2$ mole fraction and its growth rate as well as trends in atmospheric $CO_2$ mass changes. The three different analysis methods yield very similar global $CO_2$ increases from 2001 to 2020, which gives us confidence in using any one of them in climate change studies. Continuous monitoring of atmospheric $CO_2$, based on the current GAW network together with reliable global data integration methods, provides essential information. This includes understanding trends in atmospheric $CO_2$ concentra-

tion, assessing the impacts of past policies, identifying high-emission areas, informing climate models, forecasting future scenarios, and raising public awareness. Policymakers can rely on this information to support their efforts in mitigating global warming.

**Code and data availability.** All data and code necessary to calculate the global mean surface $CO_2$ mole fraction and atmospheric $CO_2$ mass is freely available from ICOS Carbon Portal (https://doi.org/10.18160/Q788-9081; Wu, 2023). The file list of results and code can be found in Sect. S4.

**Supplement.** The supplement related to this article is available online at: https://doi.org/10.5194/acp-24-1-2024-supplement. TS7

**Author contributions.** AV and ZW designed this study in discussion with YS, OT, and UK. ZW performed analysis and led the writing. YS, YN, and AO provided the GAW data and commented on the manuscript. WP and RdK provided CTE model results and relevant ObsPack data, and commented on the manuscript. XL provided NOAA data and commented on the manuscript. All authors contributed to the writing of the paper and interpretation of the results.

**Competing interests.** The contact author has declared that none of the authors has any competing interests.

**Disclaimer.** Publisher's note: Copernicus Publications remains neutral with regard to jurisdictional claims made in the text, published maps, institutional affiliations, or any other geographical representation in this paper. While Copernicus Publications makes every effort to include appropriate place names, the final responsibility lies with the authors.

**Acknowledgements.** We acknowledge Ingrid Luijkx for providing the TM5 data, WMO GAW Principal Investigators of the WMO GAW station network for providing the observational data, and Ed Dlugokencky for providing NOAA data and comments. We appreciate the support from ICOS, GAW, NOAA, and the CTE group.

**Financial support.** This research is a part of ICOS core work.

**Review statement.** This paper was edited by Christoph Gerbig and reviewed by two anonymous referees.

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

## Remarks from the language copy-editor

## Remarks from the typesetter