# Peer review of "Investigating the differences in calculating global mean surface"

_EGUsphere, 2023_

## Author Comment (AC1)

**Response to reviewer comments**

We thank reviewers for the critical comments and helpful suggestions. We have taken all these comments and suggestions into account, and they have improved our manuscript considerably. A point-by-point response to reviewers' comments please found as below.

**Responses to Referee 2:**

Wu et al. (2023) provide a valuable analysis focusing on the impact of continental site inclusion when calculating global CO2 growth rates. Employing the CTE model, the authors conduct synthetic tests to ascertain the accuracy of various growth rate estimate methods. The study is a valuable contribution to our understanding of the sampling error in the growth rate of atmospheric CO2 and is generally well-conceived. Nonetheless, the paper would benefit from clarifications and adjustments to enhance its readability and coherence.

**Main Comments:**

Presentation quality: The primary analysis of the paper focuses on the impact of including the continental sites for calculating the global CO2 growth rate. The study compares growth rate estimates from three sets of observations using, in essence, the NOAA's growth rate method:

1. NOAA: MBL sites only
2. WDCGG: MBL and some continental sites
3. CTE: MBL and a more extensive inclusion of continental sites

Given the many tests conducted and the slight variations between them, I recommend presenting this information in a table. Please specify in the table what is being compared to what is in each test to enhance the clarity of the methodology.

Response: We appreciate the reviewer for suggesting this.

The three observation networks are clarified in Table 1 (in paper or two pages below) and Fig. 1 caption (Lines 106-111): "Three observation networks are employed to assess the impact of continental site inclusion when calculating global $CO_2$ mole fraction and its growth rates. The NOAA network (43 sites, yellow stars) comprises MBL sites only. The selected GAW global network (139 sites, red dots) includes both MBL sites and continental sites, for example from the Advanced Global Atmospheric Gases Experiment (AGAGE) and European ICOS contribution network. The CTE network serves as the global network for the CTE model evaluations (230 sites, blue dots), comprises MBL sites and a more extensive inclusion of continental sites."

We have created four pair-wise comparison heatmaps, as shown in the table below, to enhance the clarity of comparisons among various networks, methodologies, periods (depending on data availability), and temporal resolutions. For example, the Figure below (Fig. 5 in the paper) displays a monthly comparison for the period 2001-2020.

|  | 2000-2020 | 1980-2020 |
|---|---|---|
| Monthly | Fig 5 (paper) | Fig S1 (supplementary) |

We have revised our manuscript to make this point clearer (Lines 216-219): "The statistical metrics assessing the agreement of these monthly comparisons are available in Fig. 5 (for 2001-2020) and Fig. S1 (for 1980-2020). The statistical metrics for the annual comparisons can be found in Fig. S2 (for 2001-2020) and Fig. S3 (for 1980-2020)."

[Figure]

**Figure 5. Pair-wise statistical metrics assess the agreement of monthly global and local $CO_2$ mole fraction (ppm) and its $G_{ATM}$ (ppm yr$^{-1}$) across various networks and methodologies (see Table 1 and Fig. 4) for the period 2001-2020. Panel (a) presents the Mean Error (ME) quantifying the difference for each pair, focusing on $CO_2$ mole fraction, while panel (b) does the same for $G_{ATM}$. The significant levels of paired t-test for ME are indicated as follows: * $p<0.1$, ** $p<0.05$, *** $p<0.01$. Panel (c) and (d) present the Root Mean Squared Error (RMSE) for $CO_2$ mole fraction and $G_{ATM}$, respectively. Panel (e) and (f) present the Pearson Correlation Coefficient (r) for $CO_2$ mole fraction and $G_{ATM}$, respectively.**

"The semi-NOAA method": The authors introduce a method called "semi-NOAA," adding unnecessary complexity to the presentation. The approach is not new, mainly the NOAA

approach on an observation set including continental sites. Referring to all the filtering and fitting procedures as components of the original NOAA method would be more effective. Subsequently, the authors could delineate any variations they are implementing compared to the standard NOAA and WDCGG methods.

Response: To avoid confusion, the method right now is named GFIT, which stands for "global fit procedure". We acknowledge that the GFIT method is not a novel approach; instead, it represents a hybrid method derived from both the standard NOAA and WDCGG methods. In the method section, we have already referred the station selection and $CO_2$ averaging method to the WDCGG approach (Text S1).

However, when it comes to the filtering, fitting, and growth rate calculation steps, we have chosen to describe and illustrate them in the method section. This decision is deliberate, as we believe it aids readers in understanding the GFIT method, particularly those who may not be familiar with the NOAA method. Without this clarity, readers might perceive these aspects as a 'black box.' Thus, our preference is to provide a detailed description and illustration of the filtering, fitting, and growth rate calculation within the method section, thereby enhancing understanding of the GFIT method.

**Minor Comments:**

1. I suggest modifying the abstract to clearly state the study's purpose: to evaluate the impact of using continental sites in CO2 growth rate calculations. It drifts off by introducing the "GFIT" method, which I do not think is the main point of this work.

   Response: We appreciate the reviewer's suggestion. We have improved the abstract.

2. It needs to be clarified how CTE is precisely used. CTE is sometimes a network, a growth rate, and a transport/inversion model run. Please use more clear terminology to differentiate. State this information in a table.

   Response: We appreciate the reviewer's suggestion. In response, we created a table (Table 1 in the paper) to differentiate and clarify the observation network and its analysis method. This table is placed in front of the results section. Specifically, CTE alone stands for CarbonTracker Europe model.

**Table 1. Description of the three observation networks and their analysis methods.**

| Terminology | Description |
|---|---|
| **NOAA network** | NOAA network comprises MBL sites only (43 sites). |
| **GAW network** | The selected GAW global network (139 sites) includes both MBL sites and some continental sites. |
| **CTE network** | The CTE network serves as the global network for the CTE model evaluations (230 sites), comprises MBL sites and a more extensive inclusion of continental sites. |
| **GAW (GFIT)** | GAW network observations analyzed using the GFIT method |
| **GAW (WDCGG)** | GAW network observations analyzed using the WDCGG method without extrapolation |

| | |
|---|---|
| **GAW (WDCGG+)** | GAW network observations analyzed using the WDCGG method with extrapolation |
| **CTE_obs (GFIT)** | CTE network observations analyzed using the GFIT method. The observations come from the ObsPack data product (Schuldt et al., 2022) |
| **CTE_output (GFIT)** | CTE model output at the 230 sites (sampled at the same location, altitude and time) analyzed using the GFIT method |
| **CTE_global (GFIT)** | CTE model output for full global grids (averaged over the first three levels, 0 to 0.35 km Alt.) analyzed using the GFIT method |
| **MLO (GFIT)** | Mauna Loa (MLO) observations analyzed using the GFIT method |
| **SPO (GFIT)** | South Pole (SPO) observations analyzed using the GFIT method |

3. The study mainly addresses monthly and multi-decadal scales. I suggest adding an analysis on annual growth rates, which have been the scales that NOAA and WDCGG report the growth rates.

   Response: In this study, we conducted analyses at both monthly and annual temporal resolutions, as we mentioned earlier. To facilitate comparisons, we created four pair-wise comparison heatmaps, as described above (also see Fig. 5, S1, S2 and S3).

4. Throughout the manuscript, excessive use of parentheses interrupts the reading flow. Consider using tables to present some of the information the reader can refer to easily.

   Response: We appreciate the reviewer's suggestion. We have incorporated the suggestions by increasing the number of tables and heatmaps to enhance the presentation, meanwhile reducing the parentheses.

5. Many sentences are unnecessarily long and could be divided into shorter, more readable sentences.

   Response: We appreciate the reviewer for the suggestion. We have reduced the number of long sentences and improved the flow and clarity.

**Technical Corrections:**

1. Line 82: The term "biased" seems unfair when referring to NOAA's estimate.

   Response: We have revised the sentence (Lines 84-86): "The NOAA estimate of global surface annual mean $CO_2$ mole fraction is expected to be lower (e.g. ~0.35 ppm lower than the WDCGG estimate, Tsutsumi et al., 2009) compared to a full global surface average because areas with large sources are not represented."

2. Lines 184-189: These lines could be made clearer to understand.

   Response: We appreciate the reviewer's comment. We have removed the mentioned lines, and instead we created a table (Table 1, as previously presented) to provide clarity regarding the observation network and its analysis methods.

We have also revised the sentence (Lines 192-194): "Global averaged surface $CO_2$ and its $G_{ATM}$ are calculated using the WDCGG method and our GFIT method based on the data from the GAW and CTE networks (Fig. 1). The different observation networks and their analysis methods are listed in Table 1."

3. Line 328: Explain the acronym IVA.

Response: We appreciate the reviewer for pointing this out. It was a typo; it should be the interannual variability (IAV). We have now corrected it.

---

## Author Comment (AC2)

**Response to reviewer comments**

We thank reviewers for the critical comments and helpful suggestions. We have taken all these comments and suggestions into account, and they have improved our manuscript considerably. A point-by-point response to reviewers' comments please found as below.

**Responses to Referee 1:**

The paper proposes an analysis of the average concentration of CO2, and its growth rate, by comparing several observation networks, and time series filtering. A comparison is also proposed with CO2 concentrations simulated by an atmospheric transport model, after a phase of assimilation of surface observations. A comparison is also made between the average concentration obtained from the surface observation network and the total amount of CO2 in the atmosphere, as simulated by the model after assimilation. Estimating these values is of course important for monitoring atmospheric radiative forcing, both for scientists and policy makers. The study therefore deserves to be published after adding few points for discussion, as detailed below. Overall, the paper is clearly written, but sometimes lacks precision and quantitative values. I think that certain recommendations, such as not extrapolating measurement series, deserve to be more clearly stipulated.

Response: We appreciate the reviewer for commenting on this. We have discussed the extrapolation done in past studies and highlighted the assumption made when extrapolating the measurement period, which could introduce uncertainty in calculating global $CO_2$ concentrations and its growth rate, in the discussion section (Lines 385-397). We have also quantified the difference between the WDCGG method with and without extrapolation, which is about 0.096 ppm in the global $CO_2$ mole fraction, although the extrapolation has a minor effect on the growth rate.

I am concerned by the fact that, every year, slightly different global CO2 estimates emerge from several networks, as detailed in this study. Even if these values differ only slightly (as the results of this study show), I think it's still a not great to multiply these slightly different estimates. Wouldn't it be possible to make a recommendation to set up a global reference network to calculate unanimously accepted values? By the way, in addition to estimates based on surface networks, every year we now also see values from networks measuring total columns measured from the ground (TCCON) or from space. This aspect is not discussed at all, but it is conceivable that these measurements could provide a more relevant assessment of the atmospheric global average. Could the model not be used to estimate this contribution from the total column measurements?

Response: We appreciate the reviewer for suggesting this.

The World Meteorological Organization has started a new initiative (Global Greenhouse Gas Watch, GGGW) that aims at establishing a reference network, that will be built on the high-quality observations already performed under the WMO GAW Program that follow consistent good practices and standards. GGGW will fill in critical geographical gaps in such a system. Setting up a global reference network for monitoring global $CO_2$ mole fraction and its growth rate is a complex and challenging task and appropriate network design will be developed as a part of the planned WMO GGGW activities. This study does not aim to provide a solution for

the global reference network, but instead it highlights the importance of station selection and analysis methodologies in monitoring global $CO_2$ mole fraction and its growth rate.

Based on the existing networks, we recommend the WMO GAW network, which includes not only the background stations (most of NOAA MBL stations) but also stations monitoring $CO_2$ sink and source from terrestrial ecosystems. We acknowledge that the current monitoring networks have uncertainties, and there is room for improvement in their design. This includes improving geographical coverage to fill observational gaps, ensuring standardization and calibration for consistency across the network, and maintaining long-term operation to increase the number of long-term running stations. Such ongoing efforts by the WMO GAW program are aligned with this goal. Additionally, the WMO has started a new initiative (Global Greenhouse Gas Watch, GGGW) that aims at establishing a reference network.

We have revised our manuscript to make this point clearer (Lines 440-446): "The high-quality observations conducted by the WMO GAW network include not only background stations (most of NOAA MBL stations) but also continental stations. This comprehensive network enables proper global average calculation. Furthermore, the WMO has initiated a new program, Global Greenhouse Gas Watch (GGGW), with the aim of establishing a reference network. This network will be built on the high-quality observations already performed under the WMO GAW program that follows consistent good practices and standards. Although the current monitoring networks have limitations in terms of geographical coverage, data consistency, and long-term measurements, they are well-equipped and have the capacity to effectively represent global surface $CO_2$ mole fraction and its growth rate and trends in atmospheric $CO_2$ mass changes."

It is well-known that the column-averaged dry air mole fraction of $CO_2$ ($xCO_2$) and surface $CO_2$ represent completely different volumes of the atmosphere and offer different coverages. Typically, $xCO_2$ is lower than surface $CO_2$, but no study has quantified this difference on a global scale. Although $xCO_2$ is not directly related to the topic of this paper, it is indeed interesting to quantify and observe this difference. Therefore, in response to the reviewer's suggestion, we conducted an analysis of TCCON and GOSAT data, comparing them to surface $CO_2$. However, please note that the analysis and results of comparing surface $CO_2$ and $xCO_2$ are not included in this paper, as we think it is beyond the scope of this study. To make this comparison meaningful, a wholly different study of the representative volumes would be required.

According to our analysis (see Fig. R1 and R2), we have found that $xCO_2$ is consistently lower than surface $CO_2$. Global TCCON (GFIT) $xCO_2$ is 0.80-1.64 ppm lower than surface $CO_2$ (i.e. data from GAW, NOAA, CTE), which is expected because the emissions originating from the surface are mixed with air with lower $CO_2$ concentration in higher levels of the troposphere. GOSAT (GFIT) $xCO_2$ has an even lower mole fraction (1.32-2.15 ppm lower), likely due to GOSAT covering a much wider area (flying between 60S and 60N) over both land and sea, and its signal goes through the atmosphere twice. In addition, the seasonality of $xCO_2$ and surface $CO_2$ are similar (r>0.99), but the seasonality changes (or amplitude) of $xCO_2$ are nearly half of those observed in surface $CO_2$. The $xCO_2$ growth rate closely matches that of surface $CO_2$, especially in case of TCCON (r=0.82-0.93, RMSE=0.16-0.26, ME=-0.01-0.06), indicating that changes in column $xCO_2$ can be well represented by surface observations.

[Figure]

**Figure R1. Comparison of globally and locally averaged $CO_2$ mole fraction (a) and its $G_{ATM}$ (b) from 1980 to 2020. Panel (a) shows the global monthly $CO_2$ mole fraction from 139 GAW sites (estimated from observations only), 43 NOAA MBL sites and those from 230 sites used in CTE (either from observations or model output). The two $xCO_2$ mole fractions are from TCCON (pink line) and GOSAT (brown line) and analysed using the GFIT method. The red and blue lines show the $CO_2$ derived from GAW (GFIT) and GAW (WDCGG), respectively. The green and orange lines show the $CO_2$ derived from CTE_obs (GFIT) and CTE_output (GFIT), respectively. The right y-axis shows their difference from NOAA $CO_2$ mole fraction, and the dashed lines show the mean of the difference over the available period. Panel (b) compares the corresponding global and local $CO_2$ growth rate, the legend refers to panel (a). The shadow area shows the uncertainty as 68% confidence interval obtained by the bootstrap analysis.**

[Figure]

**Figure R2. Pair-wise statistical metrics assess the agreement of monthly global CO₂ and xCO₂ mole fraction (ppm) and its G_ATM (ppm yr⁻¹) across various networks and methodologies (see Table 1 and Fig. 4) for the period 2010-2020. Panel (a) presents the Mean Error (ME) quantifying the difference for each pair, focusing on CO₂ mole fraction, while panel (b) does the same for G_ATM. The significance levels of paired t-test for ME are indicated as follows: * p<0.1, ** p<0.05, *** p<0.01. Panel (c) and (d) present the Root Mean Squared Error (RMSE) for CO₂ mole fraction and G_ATM, respectively. Panel (e) and (f) present the Pearson Correlation Coefficient (r) for CO₂ mole fraction and G_ATM, respectively.**

In many cases, reference is made to the CO2 concentration at Mauna Loa as a proxy for global CO2. The advantage of relying on 1 or 2 stations (MLO and SPO, for example) is that it avoids the problems of changing the configuration of the global network, and enables a fast calculation. The disadvantage is that you are stuck if the reference station fails. Having said that, I would have been interested to see a comparison of average concentrations and growth rates considering only these 2 stations.

Response: We appreciate the reviewer for this suggestion. We have added data from these stations (MLO and SPO) to our analysis (see Figure 4 below and in the paper). The data from these stations are derived from the WMO GAW and analyzed using the GFIT method. We found that the local $CO_2$ mole fraction from MLO and SPO clearly differs from the globally averaged values, and their seasonal cycles differ in timing, amplitude, and direction. Even so, the local and global growth rates behave similarly over the long-term period. However, there are evident monthly differences between the local and global growth rates, along with time shifts.

We have revised our manuscript to make this point clearer:

(Lines 245-252): "A common approach to estimate global surface $CO_2$ mole fraction is by using one or two representative sites, such as Mauna Loa (MLO) and South Pole (SPO). The globally averaged monthly surface $CO_2$ mole fractions, derived from the GAW, CTE, and NOAA networks, are significantly ($p<0.05$) lower by 0.46-0.88 ppm during 1980-2020 (Fig. S1a) and 0.45-1.19 during 2001-2020 (Fig. 5a) than the local $CO_2$ estimates solely based on MLO measurements. Conversely, these global monthly $CO_2$ mole fractions are significantly ($p<0.05$) higher by 1.91-2.24 ppm during 1980-2020 (Fig. S1a) and 2.21-2.94 during 2001-2020 (Fig. 5a) when compared to local measurements at SPO site. Furthermore, the global seasonal cycle leads the local cycle at MLO by approximately one month (estimated by averaging the time difference between the peaks of their seasonal cycles). In contrast, the local cycle at SPO is not evident and is opposite to the global seasonal cycle (Fig. 4a)."

(Lines 271-274): "Furthermore, over the long-term period of 40 years, the estimated local growth rate at MLO (ME<0.046 ppm yr$^{-1}$ higher, RMSE<0.272 ppm yr$^{-1}$, r>0.915) and SPO (ME<0.049 ppm yr$^{-1}$ lower, RMSE<0.305 ppm yr$^{-1}$, r>0.888) behaves similarly to the $G_{ATM}$ derived from GAW, CTE and NOAA network (Fig. 4b and S1). However, noticeable monthly differences between the local and global growth rates, deviating up to approximately 0.8 ppm yr$^{-1}$, and time shifts are observed (Fig. 4b)."

(Lines 407-414): "The local growth rate at MLO and SPO generally behaves similarly to the global growth rate derived from the GAW, CTE, and NOAA networks (Fig. 4b and S1). However, the local $CO_2$ mole fraction and its seasonal cycle noticeably differ from global estimates derived from different observational networks. In this regard, the utilization of individual sites for the evaluation of the global average mole fraction and its growth rate is not precise and can only be used for illustration rather than as a substitute for the proper global average calculation. The local observation sites, often situated away from significant local sources and sinks, such as MLO, provide long-term and high-quality data, serving as reference data for global $CO_2$ mole fraction. However, a single observation site cannot capture the $CO_2$

spatial variability, transport, and mixing. To overcome these limitations, global $CO_2$ trends and variations are best assessed by integrating data from multiple sources and locations."

[Figure]

**Figure 4. Comparison of globally and locally averaged $CO_2$ mole fraction (a) and its $G_{ATM}$ (b) from 1980 to 2020. Panel (a) shows the global monthly $CO_2$ mole fraction from 139 GAW sites (estimated from observations only), 43 NOAA MBL sites and those from 230 sites used in CTE (either from observations or model output). The two local $CO_2$ mole fractions are from Mauna Loa (MLO, cyan line) and South Pole (SPO, magenta line) stations, analysed using the GFIT method. The red and blue lines show the $CO_2$ derived from GAW (GFIT) and GAW (WDCGG), respectively. The green and orange lines show the $CO_2$ derived from CTE_obs (GFIT) and CTE_output (GFIT), respectively. The right y-axis shows their difference from NOAA $CO_2$ mole fraction, and the dashed lines show the mean of the difference over the available period. Panel (b) compares the corresponding global and local $CO_2$ growth rate, the legend refers to panel (a). The shadow area shows the uncertainty as 68% confidence interval obtained by the bootstrap analysis.**

Figure 4b shows a maximum divergence of methods over the last few months of 2020, which is confirmed by Table 1, where the U(Gatm) uncertainty in 2020 is about 3 times greater than in previous years. This bias is important in view of the high demand for these estimates in near-real time. You mention the problem in the discussion as a result of a side-effect of the filtering procedures. Could you propose alternative to reduce this side effect ?

Response: We appreciate the reviewer for commenting on this. We acknowledge the limitation of end-effect in the GFIT and WDCGG methods. The end-effect not only affects the last few

months but also has an influence on the first few months. For instance, there is a big shift at the beginning of 2014 in TCCON $xCO_2$ growth (Fig. R1b), which is due to two new stations joining TCCON in 2014. This implies that caution is warranted when using short-run or recently joined stations for global analysis when using the current methods (e.g. WDCGG, NOAA, and GFIT).

We have two ideas for reducing these end-effect:
a. Using machine learning to extrapolate the smoothed trend for ONE year before and after; this extrapolated portion is used exclusively for calculating local mole fraction and growth rate, BUT it is not included in the global or zonal average.
b. Similar as previous idea, the short period extrapolation is derived from bias-correcting the data comes from models (e.g. CTE) using site observations.

Overall, we think that reducing the end-effect is beyond the scope of this study. Perhaps we will consider a short follow-up paper for this.

We have revised our manuscript to make this point clearer (Lines 380-384): "Therefore, Conway et al. (1994) suggested that the growth rate curves for the last 6 months should be viewed with caution. Reducing the end-effect requires further study, such as using machine learning or bias-correction methods to extrapolate the smoothed trend for a short period (e.g. one year) before and after. This extrapolated portion is used exclusively for calculating local mole fraction and growth rate, while it is not included in the global or zonal average, as it could introduce additional uncertainty."

Data filtering: In addition to station selection, it was not clear for me if you apply a filter on the day/night periods. Mountain stations like Manua Loa are traditionally selected only during the night, while continental stations on the plains are generally selected during the day to increase the representativeness of the time series. In the discussion you mention the higher concentrations when adding continental sites, but clearly the offset will be quite strongly different if you include or not the night time CO2 accumulation in continental surface stations. Could you elaborate on this aspect ?

Response: We appreciate the reviewer for commenting on this. We agree that analyzing the $CO_2$ data during the day/night period could lead to different results. However, in this study, we are using monthly data for analysis, and in this sense, the $CO_2$ variation during the day plays a minor role in our analysis.

We have mentioned in the manuscript (Lines 144-148): "This study synchronizes monthly $CO_2$ records with the fitting and filter method developed at the NOAA Global Monitoring Laboratory (Thoning et al., 1989, Conway et al., 1994), without extrapolation. The station selection and $CO_2$ averaging method are kept the same as in the WDCGG method (Text S1). This method will be referred to as the GFIT method and will be compared to the WDCGG method without extrapolation."

Few more specific comments:

Title 'Global mean surface CO2' : For temperature measurements the elevation is normalized (e.g 10 m above ground level). This is not the cas for CO2, for which we rather avoid measuring the concentration close to the surface. Consequently It would be more accurate to refer to the 'global marine boundary layer CO2'

Response: By using the term 'Global mean surface $CO_2$,' we refer to the mean $CO_2$ concentration within the planetary boundary layer (PBL), which extends from the Earth's surface up to a few hundred/thousand meters in height. The $CO_2$ within PBL exchanges with land, vegetation, urban areas, and water. In this study, we do not solely focus on the $CO_2$ exchange with the marine boundary layer (MBL), which has the stable atmospheric conditions and interacts specifically with the ocean. Many sites in GAW and CTE network are continental (Fig. 1), thus using the term "global MBL $CO_2$" in this study is not appropriate.

We have revised our manuscript to make this point clearer (Lines 91-92): "The global mean surface $CO_2$ refers to the mean $CO_2$ mole fraction within the planetary boundary layer, which extends from the Earth's surface up to a few hundred or thousand meters in height."

Lines 47-48: please provide a reference for the conversion GtC yr-1 to ppm.yr- 1

Response: The number of the growth rate refers to Friedlingstein et al., (2022). The conversion from GtC to ppm is using the factor of 2.124, which refers to Ballantyne et al. (2012).

We have revised our manuscript to highlight this (Lines 50-51): "…, leaving a net increase of $5.0 \pm 0.2$ GtC $yr^{-1}$ of $CO_2$ in the atmosphere, corresponding to an atmospheric $CO_2$ mole fraction increase of $2.4 \pm 0.1$ ppm $yr^{-1}$ (the conversion factor comes from Ballantyne et al. (2012))."

Ballantyne, A. Á., Alden, C. Á., Miller, J. Á., Tans, P. Á. & White, J. 2012. Increase in observed net carbon dioxide uptake by land and oceans during the past 50 years. *Nature,* 488**,** 70-72. https://doi.org/10.1038/nature11299

Line 75 : " … hundreds of stations coordinated by WMO GAW: really ?"

Response: We appreciate the reviewer for noting this. Yes, the WMO GAW coordinates the surface-based observational network, which includes GAW Global (31 stations) and Regional (about 470 stations) stations. However, the number of stations currently coordinated by WMO GAW might change over time. Detailed information and the status of GAW stations can be found in the GAW Station Information System (GAWSIS, http://gawsis.meteoswiss.ch). It should be noted though that measurement programs at GAW stations can be very different and just a small percentage of these stations are involved in the measurements of greenhouse gases. The GAW program covers greenhouse gases, reactive gases, stratospheric and total ozone, aerosols, total atmospheric deposition, and solar UV radiation.

We have rephrased the sentence (Lines 78-79): "… that together form a network of hundreds of stations coordinated by WMO GAW [http://gawsis.meteoswiss.ch]."

Line 84: "…i.e. the full troposphere (up to ~8-15 km altitude) and the stratosphere or the regions of the world with substantial observational gaps"

Response: We have rephrased the sentence (Lines 86-88): "However, the two aforementioned approaches neither represent the parts of the atmosphere with low $CO_2$ mole fraction levels (i.e., the full troposphere, up to ~8-15 km altitude, and the stratosphere), nor do they cover the regions of the world with substantial observational gaps."

Line 125 : "CTE compare well…": could you more precise ?

Response: We have rephrased the sentence (Lines 131-133): "…, and the comparison of CTE $CO_2$ product to the other data assimilation systems used in GCP shows good agreement (within 0.8 ppm at all latitude bands) (Friedlingstein et al., 2022)"

Figure 4 & Line 249: Same trend with and without continental sites. Figure 4A shows that CTEobs-NOAA differences change quite markedly before and after 2000, which is not the case for GAW-NOAA scenarios. In particular, strong winter differences emerge with the development of continental stations. it's a little disturbing that this fairly clear shift between the two networks isn't reflected in long-term trends. I imagine that the difference would be seen on the trend of annual concentrations, but not when looking at annual growth rates, as this is a transient change. Could you discuss this issue ?

Response: We appreciate the reviewer for commenting on this. Yes, we can see the differences emerge with the development of continental stations when we look at the trend of annual concentrations (see Figure S4 below). The slope of the CTE trend (1.859) is clearly higher than that of the NOAA trend (1.832) due to more available continental measurements in the CTE network. The slope of the GAW (1.838) is slightly higher than the NOAA trend because GAW has more continental stations than NOAA but not as many as the CTE network (See Figure 1 in the paper).

The reason for different trends in global $CO_2$ mole fraction and global $CO_2$ growth rate is:

The global $CO_2$ mole fraction trend refers to the long-term change in the overall atmospheric $CO_2$ level over time, which is influenced by various factors, e.g. fossil fuels burning, deforestation, volcanic activity, and more. On the other hand, the global $CO_2$ growth rate trend reflects the change in the rate of $CO_2$ increase in the atmosphere, which is influenced by the balance between $CO_2$ emissions and uptake (e.g. by terrestrial ecosystem).

We have revised our manuscript to make this point clearer (Lines 275-280): "The trend analysis reveals that with development of continental sites, the slope of the trend of annual global $CO_2$ mole fraction changes from NOAA network (1.832 ± 0.029 ppm $yr^{-1}$) to CTE network (1.859 ± 0.029 ppm $yr^{-1}$) during 1980-2020 (Fig. S4). However, the $G_{ATM}$ increased steadily at a rate of 0.030 ± 0.002 ppm per year each year from 1980 to 2020 (Fig. 6a), based on the observations from the three networks (i.e. GAW, CTE and NOAA). This implies that over long-term period (here 40 years), the networks with and without continental sites exhibits the same trend of the $G_{ATM}$ and has little effect on the transient change in the rate of $CO_2$ increase in the atmosphere."

[Figure]

**Figure S4. shows the trends of global CO$_2$ mole fraction for the GAW network (red line), the CTE network (green line) and the NOAA network (black line) during the whole period 1980-2020. The cycles show the annual CO$_2$ mole fraction, respectively.**

Line 253: "red and blue lines": actually, it is red and green on Figure 5

Response: Thanks, we have corrected it in the manuscript (Lines 285).

Line 256: earlier detection of Gatm change: can you quantify how much earlier ?

Response: We appreciate the reviewer for suggesting this. We have estimated the start of CO$_2$ growth rate increase/decrease for the three strong El Niño events (i.e. 1987-1988, 1997-1998 and 2014-2016) (Table S2 below and in the paper), and further quantified the earlier detection of G$_{ATM}$ change in the observation network with more continental stations. We found that the GAW and CTE networks detect the change in CO$_2$ growth rate for the El Niño events approximately 1-2 months earlier (Table S2).

The quantification is done by smoothing the trend of CO$_2$ growth rate using a Gaussian filter (with sigma=1.96, Figure S9 below and in the paper), which aids in finding the local extrema (i.e. the start of CO$_2$ growth rate increase/decrease).

We have rephrased the sentence (Lines 282-288): "During three strong El Niño events, which are marked as grey bands in Fig. 6b, the $G_{ATM}$ derived from the GAW and CTE network (red and green lines) begins to increase approximately 1-2 months (Table S2) earlier before the El Niño events (marked as blue circles in Fig. 6b) and reaches its peak approximately 1-2 months (Table S2) earlier during the El Niño events (marked as orange circles in Fig. 6b), compared to the $G_{ATM}$ derived from the NOAA network (black line). This suggests that continental sites can aid in the early detection of $G_{ATM}$ changes resulting from changes in biogenic emission or uptake."

**Table S2. displays the estimates of $CO_2$ growth rate increase/decrease dates for the three strong El Niño events (i.e 1987-1988, 1997-1998 and 2014-2016). These estimates are calculated from the smoothed trend of $CO_2$ growth rate based on CTE, GAW and NOAA networks (Fig. S9).**

| El Niño 1987-1988 | | | | |
|---|---|---|---|---|
| | Trough ($G_{ATM}$ starts increasing) | | Peak ($G_{ATM}$ starts decreasing) | |
| **Date** | Decimal year | Days of year | Decimal year | Days of year |
| **CTE** | 1985.791635 | 289 | 1987.041665 | 15 |
| **GAW** | 1985.874965 | 319 | 1986.958295 | 350 |
| **NOAA** | 1985.874965 | 319 | 1987.124995 | 46 |
| **El Niño 1997-1998** | | | | |
| **CTE** | 1996.208325 | 76 | 1997.624975 | 228 |
| **GAW** | 1996.291655 | 106 | 1997.624975 | 228 |
| **NOAA** | 1996.374985 | 137 | 1997.708305 | 259 |
| **El Niño 2014-2016** | | | | |
| **CTE** | 2013.458315 | 167 | 2015.208325 | 76 |
| **GAW** | 2013.374985 | 137 | 2015.374985 | 137 |
| **NOAA** | 2013.541645 | 198 | 2015.374985 | 137 |

[Figure]

**Figure S9. presents the smoothed trend of $CO_2$ growth rate for each month during 1980-2020. The trends (depicted in Figure 6b) are smoothed by using a Gaussian filter (with sigma=1.96). The dots represent the local extrema, which aid in identifying the start of $CO_2$ growth rate increase/decrease.**

Line 306-307: 'The NOAA network tracks atmospheric CO2 change better': I would rather say that the result based on the NOAA netwotk comes closer to the CTE estimate.

Response: We have rephrased the sentence (Lines 349-352): "The $CO_2$ IAV based on the NOAA network exhibits a slightly closer relationship (r=0.938) with the CTE atmospheric $CO_2$

mass estimates than the GAW (r=0.861) and CTE (r=0.812) networks. This finding is consistent with the long atmospheric residence time and well-mixed nature of $CO_2$ in the NOAA network."

Lines 405-407: The conclusion need to be rephrased for clarity.

Response: We have rephrased the sentence (Lines 448-452): "Continuous monitoring of atmospheric $CO_2$, based on the current GAW network together with reliable global data integration methods, provides essential information. This includes understanding trends in atmospheric $CO_2$ concentration, assessing the impacts of past policies, identifying high-emission areas, informing climate models, forecasting future scenarios, and raising public awareness. Policymakers can rely on this information to support their efforts in mitigating the global warming."